# Coronaviruses reprogram the tRNA epitranscriptome to favor viral protein expression

Elena Muscolino [1], Mireia Puig-Torrents [1], Jaime Buigues Bisquert [2], Diogo Correa Mendonca [3], Marc Talló-Parra[1], Gemma Perez-Vilaro[1], Omar Caño-Prades[1], Gavin R. Meehan[3], Karen Kerr[3], Vanessa Herder [3], Miguel Chillón [4,5,6,7], Alfredo Castello [8], Rafael Sanjuan [2], Arvind H. Patel [3] & Juana Díez [1] ✉

Coronaviruses genomes are enriched in suboptimal A- and U-ending codons, which are typically associated with reduced translation efficiency due to limited cognate tRNA availability. How coronavirus efficiently express their proteins despite this limitation remains unclear. By analyzing their codon usage, we identify four tRNA modifications—inosine (I), queuosine (Q), 5-methylcarboxymethyluridine/ 5-methylcarboxymethyl-2-thiouridine ($mcm^5U/mcm^5s^2U$), and 5-methylcytidine/ 5-formylcytidine ($m^5C/f^5C$)—as essential for decoding their suboptimal codons. Notably, SARS-CoV-2 and HCoV-OC43 infections, representing severe and mild human infections, respectively, reprogram these modifications to favor viral protein synthesis. Mechanistically, this reprogramming was driven by altered expression of the corresponding tRNA modifying enzymes. Since both viruses induced DNA damage and oxidative stress—known to similarly alter Q, $mcm^5U/mcm^5s^2U$, and $m^5C/f^5C$ modifications to favor expression of stress response proteins—our findings support that coronavirus genomes have adapted to the tRNA modification landscape under stress conditions. Overall, coronaviruses orchestrate a codon-specific reprogramming of the host tRNA modification landscape, highlighting a conserved strategy that optimizes translation efficiency and represents a promising target for pan-coronavirus antiviral therapy development.

Coronaviruses pose a major global health threat due to their ability to cross species barriers and establish infections in humans[1]. These zoonotic transmissions, exemplified by the COVID-19 pandemic, can give rise to novel viral strains with unpredictable consequences[2]. Gaining insights into how coronaviruses exploit host cell machineries to multiply is crucial for the development of novel antiviral approaches. A key strategy involves their exploitation of the host translation machinery. The genome of coronaviruses consists of a single-stranded, positive-sense (+)RNA, ranging from 26 to 30 Kb, that encodes for 23 to 32 viral proteins. Upon entry into the cytoplasm, these proteins are

[1]Molecular Virology group, Department of Medicine and Life Sciences, Universitat Pompeu Fabra, Barcelona, Spain. [2]Instituto de Biología Integrativa de Sistemas (I2SysBio), Parc Cientific de la Universitat de València, Valencia, Spain. [3]CVR-CRUSH, MRC-University of Glasgow Centre for Virus Research, Sir Michael Stoker Building, Garscube Campus, Glasgow, United Kingdom. [4]Department of Biochemistry and Molecular Biology, Institut de Neurociènces (INc), Universitat Autònoma de Barcelona, Barcelona, Spain. [5]Vall d'Hebron Institut de Recerca (VHIR), Barcelona, Spain. [6]Unitat Producció de Vectors (UPV), Universitat Autònoma de Barcelona, Barcelona, Spain. [7]Institució Catalana de Recerca i Estudis Avançats (ICREA), Barcelona, Spain. [8]MRC-University of Glasgow Centre for Virus Research, Sir Michael Stoker Building, Garscube Campus, Glasgow, United Kingdom. ✉e-mail: juana.diez@upf.edu

directly translated from the genomic (+)RNA or from subgenomic (+) RNAs transcribed during infection[3]. Their efficient and timely expression is essential for viral replication and expansion. For this, coronaviruses, like all viruses, rely entirely on the host translation machinery[4]. Despite this reliance, their genomes are enriched in suboptimal codons and consequently should be translated with poor efficiency. The redundancy of the genetic code allows multiple codons (synonymous codons) to code for the same amino acid. However, the usage of these synonymous codons is not random; rather, it is universally biased and organism-specific. In mammals, highly expressed genes preferentially use G- and C-ending codons, whose cognate tRNAs are highly abundant, whereas coronavirus genomes are enriched in A- and U-ending codons, whose cognate tRNAs are scarce[5,6]. This codon bias is predicted to slow down translation elongation and reduce protein expression efficiency. How coronaviruses achieve efficient genome translation in mammals despite their suboptimal codon usage remains an unresolved and fundamental question.

The redundancy of the genetic code arises through two key mechanisms. First, isoacceptor tRNAs, distinct tRNA molecules that carry the same amino acid but have different anticodons, enable the recognition of different synonymous codons. For example, the six synonymous codons specifying Leucine (Leu) are recognized by distinct tRNA isoacceptors. Second, some tRNA molecules exhibit flexibility in codon recognition, allowing a single tRNA to pair with multiple codons. This flexibility primarily results from wobble base pairing between the third position of the codon in the mRNA and the first position of the anticodon in the tRNA (N34), known as the wobble position[7]. tRNAs deliver the amino acids to the ribosome by pairing their anticodons (located at positions 34, 35 and 36) with complementary codons in mRNAs (positions 3, 2, and 1, respectively). While positions 1 and 2 follow strict Watson-Crick base pairing rules, the third position allows for non-standard base pairing, which can be further regulated by chemical modifications at position 34[8]. For example, uridine at position 34 (U34) can pair with both adenine (A) and guanine (G). Chemical modifications at U34 modulate this flexibility, either enhancing or restricting non-canonical base pairing.

tRNA levels and functions are regulated by a wide array of chemical modifications collectively known as the tRNA epitranscriptome. Position 34 is a hotspot for these modifications, hosting a wide range of chemical modifications that ultimately enable a limited set of anticodons to efficiently decode the synonymous codons[9]. Importantly, tRNA modifications are dynamically regulated in response to environmental changes and cellular stress conditions, allowing cells to fine-tune translation efficiency and adapt to fluctuating metabolic demands. We have recently shown that infection with chikungunya virus (CHIKV), another (+)RNA virus with an A-ending codon bias, triggers a DNA damage response that ultimately leads to the overexpression of KIAA1456, a tRNA-modifying enzyme. This enzyme catalyzes the mcm$^5$U modification at U34, favoring the decoding of the suboptimal Glutamic acid (Glu) GAA, Lysin (Lys) AAA, Glutamine (Gln) CAA, and Arginine (Arg) AGA codons over the optimal G-ending counterparts[10]. These changes reprogram codon optimality, selectively enhancing the translation of DNA damage stress response genes and the CHIKV RNA genome, both highly enriched in these A-ending codons. Hence, CHIKV codon usage optimally aligns with the reprogrammed tRNA modification landscape induced by the stress response to efficiently express the stress response proteins. Whether this strategy is also employed by coronaviruses and whether additional tRNA modifications might influence coronavirus protein expression are unknown. To address these fundamental questions, here we conducted a comprehensive analysis of the interplay of coronaviruses with the host tRNAome. Our approach combined bioinformatic analyses of coronavirus codon usage in the context of host tRNA modifications with Liquid Chromatography–Tandem Mass Spectrometry (LC–MS/MS) and modification-induced misincorporation tRNA sequencing

(mim-tRNAseq) analysis to profile the tRNA epitranscriptome and tRNA abundance in SARS-CoV-2 and HCoV-OC43 infected cells, as representatives of high- and low-pathogenic human coronaviruses, respectively.

We found that SARS-CoV-2 and HCoV-OC43 infections reprogram the tRNA epitranscriptome to enhance viral protein expression by altering the levels of four tRNA modifications that affect the decoding of more than half of their enriched codons. Mechanistically, these changes correlate with parallel shifts in the expression of the corresponding tRNA-modifying enzymes. Remarkably, these codons are highly conserved across the coronavirus genera.

## Results

### Coronavirus genomes are enriched in codons whose decoding is influenced by tRNA modifications

Among the four coronavirus genera, only *Alphacoronavirus* and *Betacoronavirus* include species that infect humans. The *Alphacoronavirus* genus comprises HCoV-229E and HCoV-NL63, both associated with mild infections. In the *Betacoronavirus* genus, HCoV-OC43 and HCoV-HKU1 also cause mild illnesses, whereas SARS-CoV, MERS-CoV and SARS-CoV-2 are highly pathogenic. To systematically analyze the codon usage of *Alphacoronaviruses* and *Betacoronaviruses*, we first examined the relative synonymous codon usage (RSCU), grouping them by their host species and, separately, focusing solely on the human-infecting ones (Fig. 1A–C). Both *Alphacoronaviruses* and *Betacoronaviruses*, including the human-infecting group, were similarly enriched in suboptimal A- and U-ending codons (Fig. 1A–C) over G- and C-ending codons, with the Leu-UUG codon being a notable exception. Three additional features stood out. First, as previously observed in human- and bat-infecting coronaviruses[11], there is a distinct preference for U-ending over A-ending codons throughout all coronaviruses, particularly in comparison to other human-infecting (+)RNA virus genera (Supplementary Fig. 1). Second, certain specific U ending codons, such as Proline (Pro) CCU, Arg-CGU, Alanine (Ala) GCU, Glycine (Gly) GGU, Valine (Val) GUU and Serine (Ser) UCU, are particularly enriched across coronaviruses. Third, within human-infecting coronaviruses, low-pathogenic strains exhibited a higher enrichment in some U-ending codons (i.e., Gly-GGU, Phenylalanine (Phe) UUU, Ser-AGU, Val-GUU) and in the Leu-UUG codon than high-pathogenic strains (Supplementary Fig. 1). Next, we focused on human-infecting coronaviruses to analyze codon bias from an amino acid perspective by comparing the synonymous codons (Fig. 1D). For simplicity, human-infecting coronaviruses will be referred to as coronaviruses hereafter.

We evaluated how many codons enriched in the coronavirus genome might be influenced by modifications that occur at the 34 wobble position of tRNAs. Indeed, the decoding of 48% of codons enriched in coronavirus genomes is affected by four tRNA modifications: I, Q, mcm$^5$/mcm$^5$s$^2$U, which occur exclusively at position N34, and m$^5$C, which can also occur at other tRNA positions besides N34. The I modification hinders the decoding of eight U-ending codons (Ala-GCU, Arg-CGU, Isoleucine (Ile) AUU, Leu-CUU, Pro-CCU, Ser-UCU, Threonine (Thr) ACU, or Val-GUU), which are among the most highly enriched throughout coronaviruses (Fig. 1D). When unmodified, the corresponding tRNAs decode only the U-ending codons. However, when I-modified, the synonymous A- and C-ending codons are also decoded, as I base pairs with U, C, or A (Fig. 1E)[12]. This expansion of decoding capacity would disfavor the decoding of coronavirus codons. The Q modification and its derivatives, mannosyl-Q (manQ) and galactosyl-Q (galQ), favor the decoding of four U-ending codons (Asparagine (Asn) AAU, Aspartic acid (Asp) GAU, Histidine (His) CAU, and Tyrosine (Tyr) UAU), with galQ found in human tRNA$^{Tyr}$, and manQ found in human tRNA$^{Asp}$ (Fig. 1D)[13]. Since the corresponding tRNAs are absent in humans, their decoding relies on Q modifications at G34 in the anticodon loop of tRNAs decoding their synonymous C-ending

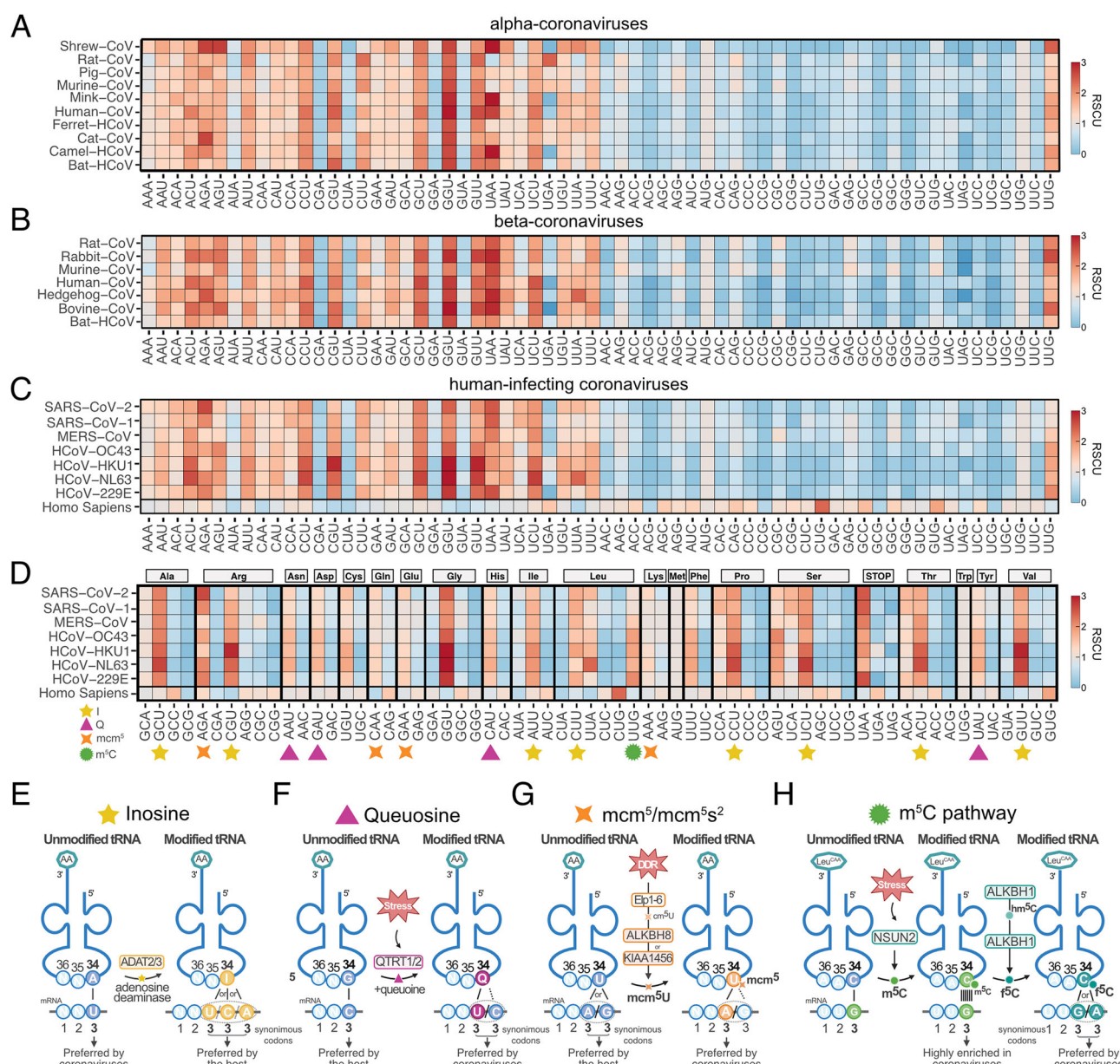

**Fig. 1 | Relative Synonymous Codon Usage (RSCU) analysis of coronavirus sequences. A, B** RSCU analysis of coding sequences of key genes from *Alphacoronavirus* and *Betacoronavirus* and of *Homo sapiens*. The coronavirus genes included ORF1ab, spike, envelope, membrane protein, and nucleocapsid genes, concatenating these sequences in the aforementioned order. Genes such as 3a/b, 4, and 7b were excluded from the study due to annotation inconsistencies. The RSCUs were grouped by their respective hosts. **C** RSCU analysis of human-infecting coronaviruses. **D** RSCU of human-infecting coronaviruses organized by synonymous codons. Codons that are modified either by I (yellow star), Q (triangle) mcm⁵U/mcm⁵s²U (orange star) or m⁵C (green circle) are indicated. **E**–**H** Graphical representation of I, Q, mcm⁵U/mcm⁵s²U and m⁵C modification pathways. Panels (**E**–**H**) were created in Biorender. *Diez, J. (2026)* https://BioRender.com/9crp7a7.

codons[14,15]. When Q-modified, G34, which normally pairs with C or with near-cognate Q-decoded codons (such as Glu-GAA and Glu-GAG), can also base pair with U, favoring decoding of coronavirus codons (Fig. 1F). The mcm⁵U/ mcm⁵s²U modifications favor the decoding of four A-ending codons (Arg-AGA, Gln-CAA, Lys-AAA and Glu-GAA) over the synonymous G-ending counterparts. When unmodified, U34 pairs with both A and G due to wobble pairing. However, when modified, pairing with A is favored, thereby enhancing decoding of coronavirus codons (Fig. 1G)[10,16,17]. Altogether, the decoding of 75% of the U-ending codons and 42% of the A-ending codons enriched in coronaviruses are influenced by tRNA modifications at position 34. Finally, efficient decoding of Leu-UUG, the only G-ending codon exceptionally enriched in coronavirus genomes, is favored by the tRNA modification m⁵C and its oxidized forms 5-hydroxymethylcytosine (hm⁵C) and f⁵C when

added at position C34 in the corresponding tRNA[18,19]. In addition, f⁵C modifications also expand the tRNA recognition to Leu-UUA, which is also enriched in coronaviruses[20].

While I34 modification has been shown to be very stable, the levels of Q, mcm⁵U/mcm⁵s²U and m⁵C/hm⁵C/f⁵C modifications are upregulated under stress conditions[18,21–23]. This coordinated interplay between the tRNA epitranscriptome and stress responses allows the rapid and coordinated expression of stress-response proteins, which are enriched in codons whose decoding is favored by these modifications. Specifically, Q, mcm⁵U/mcm⁵s²U, and m⁵C have been reported to be overexpressed in response to DNA damage (DDR) (mcm⁵U/mcm⁵s²U)[24] and oxidative stress responses (all)[18,21,25]. This suggests that coronavirus codon usage has evolved to exploit the altered tRNA epitranscriptome triggered by virus-induced stress

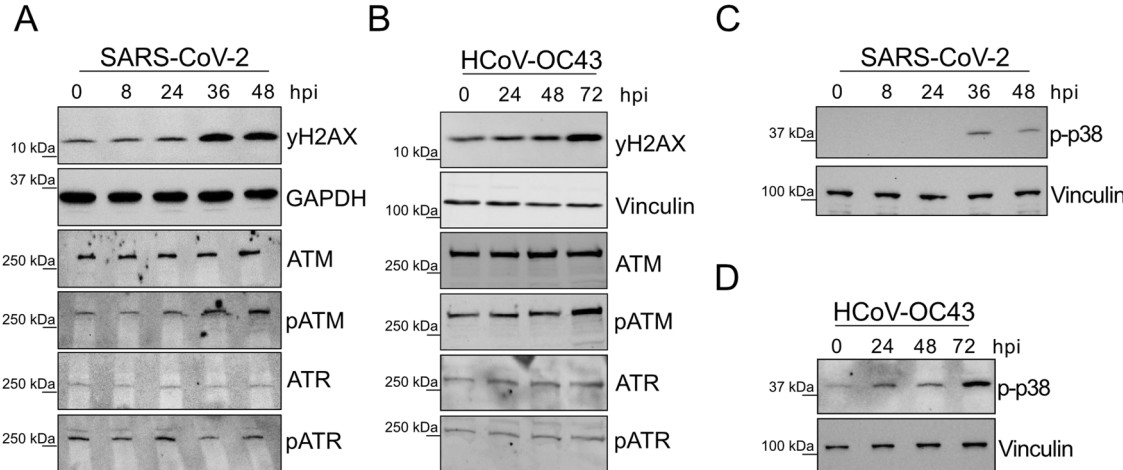

**Fig. 2 | Coronaviruses trigger DNA Damage Response (DDR) and oxidative stress. A, B** Western blot analysis of DDR proteins γH2AX, ATM, ATR, pATM and pATR in SARS-CoV-2 infected Calu3 cells and HCoV-OC43 infected MRC5 cells at several hpi. Representative example of three independent replicates (*n* = 3 where n denotes the number of biological replicates, as used throughout the manuscript). **C, D** Western blot analysis of p-p38 protein levels in SARS-CoV-2 infected Calu3 cells and HCoV-OC43 infected MRC5 cells at several hpi (*n* = 3). Vinculin and GAPDH were used as loading controls.

responses. To test whether coronavirus infection triggers stress responses, we used SARS-CoV-2 and HCoV-OC43 as models of high- and low-pathogenic human coronaviruses and analyzed DNA damage-associated protein levels and oxidative stress markers. Hereafter, SARS-CoV-2 infections were conducted in Calu3 lung epithelial cells, an established model for studying SARS-CoV-2 infection due to their high expression of Angiotensin-Converting Enzyme 2 (ACE2) receptors, facilitating efficient viral entry and replication. For HCoV-OC43, MRC5 lung fibroblasts were chosen because they support the replication of low-pathogenic coronaviruses. In both infections, we observed increased DNA damage and oxidative stress, as indicated by the upregulation of key stress biomarkers detected via western blot (Fig. 2A, B). In the DNA damage stress response, the Ataxia-Telangiectasia Mutated (ATM) or/and the Ataxia-Telangiectasia and Rad3-related (ATR) are activated, leading to the phosphorylation of various substrates, including the H2AX histone variant (γH2AX) and the kinases themselves. Notably, γH2AX can also be phosphorylated in response to certain oxidative stress conditions[26]. We observed that γH2AX was upregulated in both infection conditions, beginning at 36 hours post-infection (hpi) for SARS-CoV-2 and between 48 and 72 hpi for HCoV-OC43. Further analysis revealed phosphorylation of ATM, but not ATR, under both coronavirus infections (Fig. 2A, B), as shown previously in SARS-CoV-2 infected h2h7 cells[27]. In the oxidative stress response, the p38 mitogen-activated protein kinase (p38 MAPK) plays a key regulatory role. The phosphorylated form of this kinase (p-p38) is considered a biomarker of oxidative stress and other stressors such as UV irradiation and inflammation[28]. Both SARS-CoV-2 and HCoV-OC43 infections induced p38 phosphorylation, starting at 36 hpi for SARS-CoV-2 and 24 hpi for HCoV-OC43 (Fig. 2C, D). Furthermore, ROS production was confirmed by directly measuring ROS levels in infected cell (Supplementary Fig. 2A, B). These findings suggest that coronaviruses have adapted their codon usage to thrive in a tRNA environment in which stress responses are activated. Consistent with this, previous studies have shown that inhibitors of the DNA damage stress response pathway and antioxidant impair coronaviruses infection[29–33]. Furthermore, inhibitors of the DNA damage stress response reduce both p-p38 levels and coronavirus protein expression (Supplementary Fig. 2C). Collectively, our analyses uncover novel features of coronavirus codon usage and suggest that these may have evolved, at least in part, in response to altered host tRNA environments triggered by coronavirus-induced stress responses. To further address this fundamental issue, we comprehensively analyzed alterations in the tRNA landscape during coronavirus infection, focusing on both tRNA modifications and abundance.

## Coronaviruses alter the host tRNA epitranscriptome in vitro and in vivo

To investigate alterations in tRNA modifications during coronavirus infection, and to determine the optimaltime point when the viruses were fully replicating while the cells remained viable, we first analyzed the infection kinetics of SARS-CoV-2 in Calu3 cells and of HCoV-OC43 in MRC5 cells (Supplementary Fig. 3). For SARS-CoV-2, this time point was 32 hpi (Supplementary Fig. 3A–C), whereas for HCoV-OC43, it was 48 hpi (Supplementary Fig. 3D–F). At these time points, tRNAs were extracted from non-infected and coronavirus-infected cells and analyzed using LC-MS/MS to identify and quantify tRNA modifications (Fig. 3 and Supplementary Data 1). Both coronaviruses induced statistically significant changes in the tRNA epitranscriptome, as indicated by the log2 fold changes between infected and non-infected cells (Fig. 3A). In SARS-CoV-2 infected cells, these changes included three tRNA modifications at position 34 in the anticodon loop (Fig. 3A, black stars). In HCoV-OC43 infected cells, five tRNA modifications at position 34 of the anticodon loop (Fig. 3A, black stars), along with four additional modifications at other tRNA residues (Fig. 3A, yellow stars). Three modifications at position 34 exhibited the same altered pattern in both viruses: down regulation of I and f⁵C and upregulation of mcm⁵U (Fig. 3B–D). However, in SARS-CoV-2-infected cells, significant changes were observed in 2′-O-methyl-inosine (Iᵐ) instead of I. While I and its derivative Iᵐ appear to share similar functions, Iᵐ is less characterized and has been implicated in tRNA stability[34]. The observed down regulation of I/Iᵐ and upregulation of mcm⁵U favor the decoding of U- (Fig. 1D, E) and A-ending codons (Fig. 1D, G) that are highly enriched in coronaviruses. mcm⁵U can undergo further modification to mcm⁵s²U in tRNAs; however, its detection by LC-MS/MS is challenging due to its instability. Although statistically significant changes in mcm⁵s²U levels were not detected at the time of analysis, we observed its upregulation at 48 hpi post-SARS-CoV-2 infection, suggesting a transition from the mcm⁵U to mcm⁵s²U at later time points (Supplementary Fig. 4A). Importantly, mcm⁵U upregulation was also observed in vivo in Syrian hamsters infected with SARS-CoV-2. Three female and three male Syrian hamsters were infected, and two days post-infection, the upper right lobe of the right lung was collected for tRNAs analyses. Three uninfected females and three uninfected males were used as mock-infected controls (Fig. 3E). While qPCR analysis of SARS-CoV-2

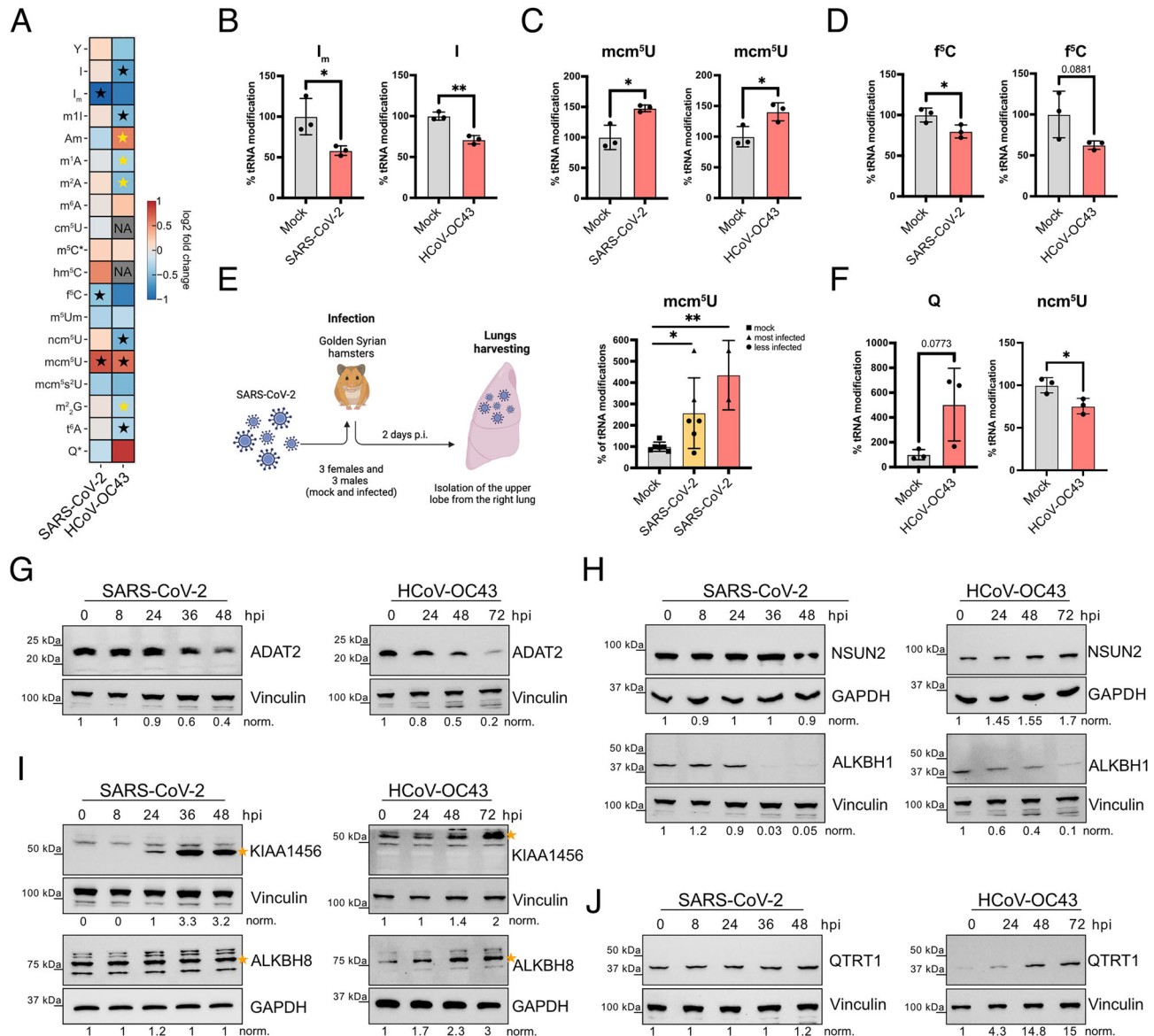

**Fig. 3 | LC-MS/MS analysis of tRNA epitranscriptome landscapes under SARS-CoV-2 and HCoV-OC43 infection conditions. A** Heatmap showing log2 fold changes of tRNA modifications in SARS-CoV-2–infected Calu3 cells (MOI 3, 32 hpi) relative to non-infected cells, and in HCoV-OC43–infected MRC5 cells (MOI 0.1, 48 hpi) relative to non-infected cells, as measured by LC–MS/MS (*n* = 3). Values represent the mean log2 fold change. Statistically significant modifications located in the anticodon loop are indicated by black stars, whereas significant modifications at other tRNA positions are indicated by yellow stars. Statistical significance was assessed using a two-sided, nonparametric Mann–Whitney U test (unpaired). **B**–**D** Statistical analysis of tRNA modifications shown in (**A**) was performed using a two-sided, nonparametric Mann–Whitney U test (unpaired; *n* = 3). Exact *p*-values are reported: Im, *p* = 0.0347 (SARS-CoV-2) and *p* = 0.0021 (HCoV-OC43); mcm⁵U, *p* = 0.0160 (SARS-CoV-2) and *p* = 0.0345 (HCoV-OC43); f⁵C, *p* = 0.0403 (SARS-CoV-2) and *p* = 0.0881 (HCoV-OC43). Data are presented as mean ± s.d. Individual data points are shown. **E** Schematic representation of the SARS-CoV-2 infection in the hamster model and quantification of mcm⁵U levels in hamster lungs by LC–MS/MS. Animals were infected with SARS-CoV-2 carrying the D614G mutation and stratified

into all infected animals (yellow; *n* = 6) or highly infected animals (red; *n* = 2), based on viral load. Differences relative to mock-infected controls were assessed using a two-sided, nonparametric Mann–Whitney U test. Exact *p*-values are *p* = 0.0436 for all infected animals versus mock and *p* = 0.0010 for highly infected animals versus mock. Data are presented as mean ± s.d. Individual data points are shown. **F** Statistical analysis of ncm⁵U and queuosine (Q) modifications measured by LC–MS/MS in HCoV-OC43–infected cells compared to mock-infected controls. Measurements were performed at 48 hpi for ncm⁵U and at 72 hpifor Q (*n* = 3). Differences between infected and mock-infected conditions were assessed using a two-sided, nonparametric Mann–Whitney U test. Exact *p*-values are *p* = 0.0292 for ncm⁵U and *p* = 0.0773 for Q. Data are presented as mean ± s.d. Individual data points are shown. **G**–**J** Western blot analysis of tRNA modifying enzymes (ADAT2, KIAA1456 (indicate by a star), ALKBH8 (indicate by a star), ALKBH1, NSUN2, QTRT1) associated with tRNA modifications in SARS-CoV-2 infected Calu3 cells (MOI 3; 0, 8, 24, 36, and 48 hpi) and HCoV-OC43 infected MRC5 cells (MOI 0.1; 0, 24, 48, and 72 hpi, *n* = 3) GAPDH and Vinculin have been used as loading controls. Panel (**E**) was created in Biorender. Diez, J. (2026) https://BioRender.com/24heq5e.

RNA revealed variability in infection levels among the samples (Supplementary Fig. 4B), LC–MS/MS analysis of tRNA modifications showed a consistent increase in mcm⁵U levels in all samples, particularly in those that were the most highly infected (Fig. 3E). Lastly, f⁵C downregulation is predicted to hinder the decoding of Leu-UUG/UUA,

enriched in coronaviruses (Fig. 1D, H). However, the precursors m⁵C, which stabilize the interaction with Leu-UUG codon, exhibit an opposite trend, potentially compensating for f⁵C loss. In addition, m⁵C/hm⁵C/f⁵C modifications also occur in mitochondrial tRNA-iMet, where they facilitate recognition of the Ile-AUA codon as methionine,

ensuring efficient mitochondrial translation initiation[19] (Supplementary Fig. 4C). Given that f[5]C is present in both cytoplasmic and mitochondrial tRNAs, and LC-MS/MS analysis cannot discriminate between cytosolic and mitochondrial tRNA pools, the observed changes may reflect mitochondrial stress responses rather than direct effects on viral protein translation.

Among the six statistically significant tRNA modification changes detected exclusively in HCoV-OC43-infected cells, two were located at position N34, while the others occurred at other tRNA positions (Fig. 3A, F and Supplementary Fig. 4D). Modification changes at position N34 included the upregulation of Q and the downregulation of 5-carbamoylmethyluridine (ncm[5]U) (Fig. 3F). The upregulation of Q aligns with coronavirus codon usage, as this modification is required to decode a subset of U-ending codons that are particularly enriched in coronaviruses (Fig. 1D). Recent studies suggest that Q modification, as f[5]C, may also occur in mitochondrial tRNAs[35,36]. Studies in yeast have shown that the ncm[5]U modification is added to U34 of tRNAs decoding Ala-GCA, Pro-CCA and Val-GUA to ensure proper decoding. Although these codons are present in coronavirus genomes, they are less enriched compared to their synonymous U-ending counterparts. Since the formation of mcm[5]U and ncm[5]U derives from the same precursor (Supplementary Fig. 4E), it is not unexpected that an increase in mcm[5]U coincides with a decrease in ncm[5]U. For tRNA modification changes not located at position N34, we observed an upregulation of 2′-O-methyladenosine (mA), a modification that enhances tRNA stability, and a down regulation of N[1]-methyladenosine (m[1]A), N[2]-methyladenosine (m[2]A), N[1]-methylinosine (m[1]I) and N6-threonylcarbamoyladenosine (t[6]A). These modifications are involved in tRNA folding and stability (m[1]A, t[6]A), recognition by aminoacyl-tRNA synthetases and ribosomes (m[1]A, t[6]A) and codon-anticodon interaction (m[2]A, m[1]I, t[6]A) (Supplementary Fig. 4D). As some of these modifications have not yet been assigned to specific tRNAs or may modify the same tRNA molecule in multiple nucleotide positions, it remains challenging to determine their positive or negative effects on coronavirus protein expression.

Collectively, our findings indicate that the alterations in tRNA modifications at the wobble position induced by SARS-CoV-2 and HCoV-OC43 infections enhance the decoding of codons that are highly enriched in their genomes, potentially conferring a translational advantage over host codons.

## Coronavirus-induced tRNA modification changes are driven by altered expression of tRNA modifying enzymes

To gain insights into the mechanisms driving these alterations, we assessed whether changes in tRNA modifications correlated with changes in the expression of their corresponding modifying enzymes. Protein levels of these enzymes were examined in Calu3 and MRC5 cells infected with SARS-CoV-2 and HCoV-OC43, respectively, at various time points using western blot (Fig. 3G–J). Consistent with the LC-MS/MS-detected reduction in I/I[m] modification levels in tRNAs under both infection conditions, ADAT2 expression, a key component of the ADAT2/ADAT3 complex responsible for I modification, was downregulated[37]. ADAT2 protein levels declined from 36hpi for SARS-CoV-2 and 48 hpi for HCoV-OC43 (Fig. 3G). The f[5]C modification pathway begins with the methylation of C34 to m[5]C, catalyzed by NSUN3 in mitochondrial tRNA-iMet and by NSUN2 in cytoplasmic tRNA[Leu][CAA] under stress conditions. This is followed by the oxidation of m[5]C to hm[5]C and subsequently to f[5]C, mediated by ALKBH1[19]. Here after, we used the standard nomenclature for tRNAs, which denotes the amino acid and the anticodon. For example, tRNA[Leu][CAA] recognizes the Leu-UUG codon. NSUN2 levels remained unchanged during SARS-CoV-2 infection but moderately increased in HCoV-OC43 infection from 48 hpi onwards (Fig. 3H). However, ALKBH1 expression was vastly downregulated in both infections (Fig. 3H), aligning with LC-MS/MS data showing a marked reduction in f[5]C modifications (Fig. 3C). A

similar correlation was observed between the upregulation of mcm[5]U/mcm[5]s[2]U and Q tRNA modifications and the corresponding upregulation of their modifying enzymes. The mcm[5]U/mcm[5]s[2]U modification is mediated by KIAA1456 or its paralog ALKBH8 (Fig. 1G)[10,38–40]. Upon SARS-CoV-2 infection, KIAA1456 expression increased, while both KIAA1456 and ALKBH8 were upregulated during HCoV-OC43 infection (Fig. 3I). Q modification is catalyzed by the QTRT1/QTRT2 enzyme complex[41,42]. When quantifying QTRT1 expression, a strong upregulation was observed in HCoV-OC43 but not in SARS-CoV-2 infected cells (Fig. 3J), consistent with the LC-MS/MS results (Fig. 3A).

To establish a causal link between the alteration of these enzymes and viral protein expression, we manipulated the expression of selected tRNA modifying enzymes. These experiments were performed using HCoV-OC43 infection in A549 cells because the observed changes in tRNA modifications were conserved across both viruses analyzed, transfection efficiency in Calu3 and MRC5 cell lines (used for SARS-CoV-2 and HCoV-OC43 infection, respectively) is poor, and A549 cells support efficient HCoV-OC43 replication and transfection (Supplementary Fig. 3D, F). Consistent with a functional link between tRNA epitranscriptome remodeling and viral protein production, siRNA-mediated knockdown of ELP3 (mcm[5]U modification pathway) and QTRT1 (queuosine modification pathway) significantly reduced viral protein levels (Fig. 4A, B, D, E). ELP3 was selected for silencing because it acts upstream of KIAA1456 and ALKBH8 in the mcm[5]U modification pathway and is strongly upregulated during HCoV-OC43 infection (Fig. 4A). Similarly, overexpression of ADAT2 (A-to-I modification enzyme) or ALKBH1 (f[5]C pathway) also led to a pronounced decrease in viral protein expression (Fig. 4G, H). In contrast, NSUN2 silencing had no detectable effect (Fig. 4C, F). This lack of effect, together with the observed downregulation of ALKBH1 during infection (Fig. 3H) and the strong inhibition of viral protein production upon its overexpression, suggests that viral infection primarily modulates the mitochondrial NSUN3–ALKBH1 pathway rather than the cytosolic NSUN2-dependent m[5]C modification. This interpretation is further supported by the absence of significant changes in global tRNA m[5]C levels analyzed by LC-MS/MS during infection (Fig. 3A). Together, these findings demonstrate that alterations in mcm[5]U, Q, and I tRNA-modifying enzymes during infection not only align with viral codon usage but also functionally contribute to coronavirus protein production.

To further confirm the link between coronavirus infection, tRNA modifications and viral protein expression, we analyzed codon frequency in host transcripts translation-repressed (TR) versus translation-activated (TA) in ribosome profiling datasets from SARS-CoV-2–infected Calu3 cells[43]. We reasoned that coronavirus-induced alterations in tRNA modifications would modify codon optimality, thereby influencing not only viral but also host mRNA translation. Consistently, we found that, similar to coronavirus mRNAs, TR mRNAs were enriched in codons preferentially decoded by I-modified tRNAs (Supplementary Fig. 5A–C). In contrast, host TA mRNAs were enriched in codons preferentially decoded by tRNAs bearing I-unmodified, mcm[5]U and Q modifications. Moreover, ribosome occupancy analyses revealed reduced occupancy at these codons in line with enhanced translation elongation during infection (Supplementary Fig. 5D–F).

Interestingly, tRNA-modifying enzymes responsible for introducing the mcm[5]U/mcm[5]s[2]U modification are enriched in codons that improve their own decoding efficiency[44]. This creates a feedback loop that amplifies or reduces their expression in response to changes in their activity. To study if this phenomenon extends to other tRNA modifying enzymes, we analyzed the RSCU of the human ADAT2 (Fig. 5A, B), QTRT1 (Fig. 5C), KIAA1456/ALKBH8 (Fig. 5D), and NSUN2 (Fig. 5E) enzymes and of related proteins from the same pathways. In all cases except QTRT1/QTRT2, we found an enrichment in the codons whose translation they enhance.

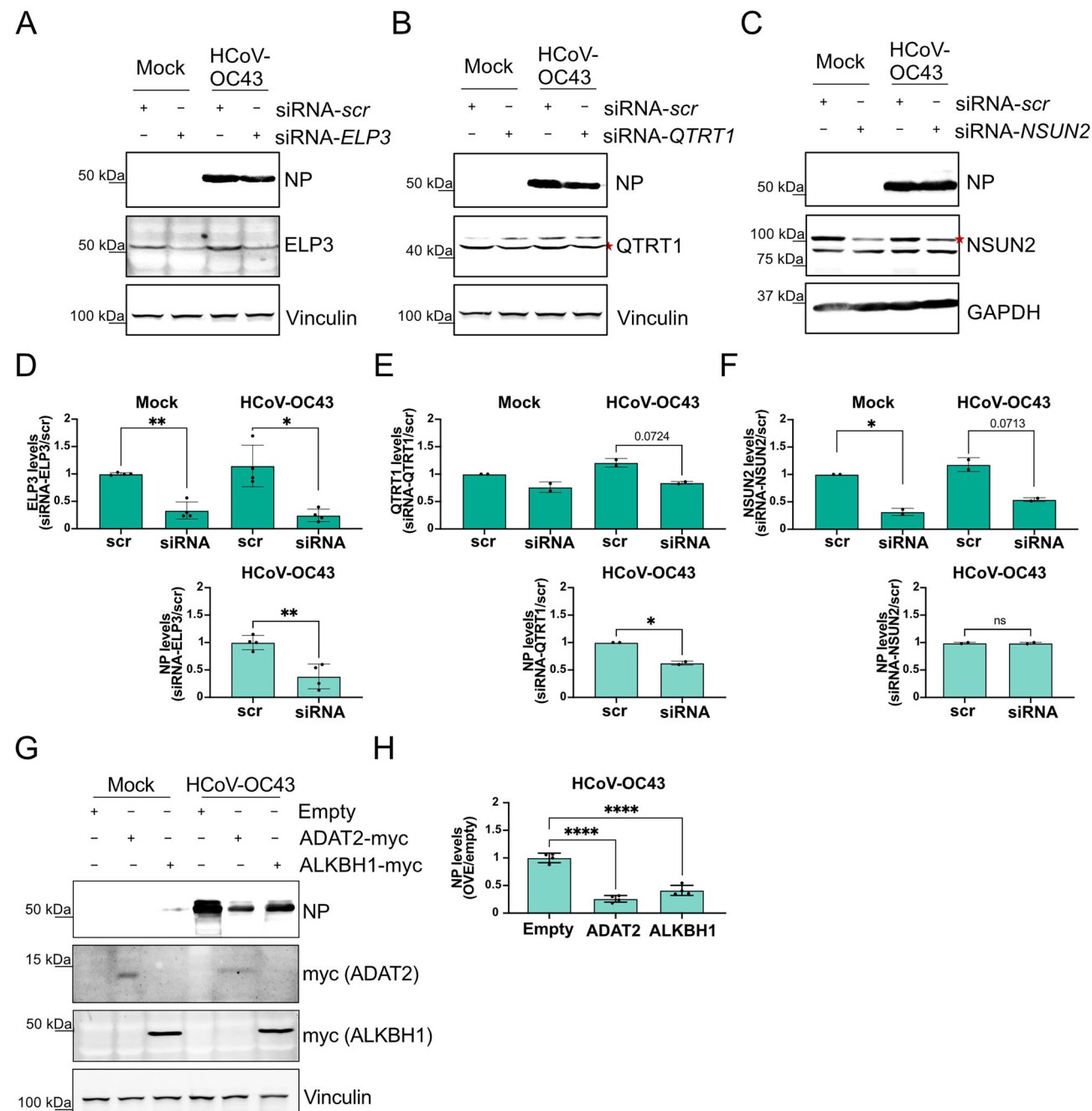

**Fig. 4 | Functional impact of tRNA-modifying enzymes in coronavirus infection.**
**A**–**C** Representative western blots showing siRNA-mediated knockdown of ELP3 (mcm⁵U/mcm⁵s ≤ U pathway), QTRT1 (Q pathway), and NSUN2 (m⁵C pathway) in HCoV-OC43–infected A549 cells (MOI 0.1). HCoV-OC43 N protein (NP) levels were measured by western blot at 48 hpi. A non-targeting siRNA (siRNA-scr) was used as a control. **D**–**F** Quantification of ELP3, QTRT1, NSUN2, and viral NP protein levels in siRNA-treated cells relative to non-targeting siRNA (scr) controls. Vinculin and GAPDH were used as loading controls. Measurements were obtained from independent biological replicates ($n = 4$ for panel D; $n = 2$ for panels **E** and **F**). Statistical significance was assessed using a two-sided, nonparametric Mann–Whitney U test (unpaired). **D** ELP3 levels: $p = 0.003$ for mock siRNA-scr vs mock siRNA-ELP3; $p = 0.0137$ for HCoV-OC43 siRNA-scr vs HCoV-OC43 siRNA-ELP3; NP levels: $p = 0.0057$ for HCoV-OC43 siRNA-scr vs HCoV-OC43 siRNA-ELP3. **E** QTRT1 levels: $p = 0.17$ for mock siRNA-scr vs mock siRNA-QTRT1; $p = 0.0724$ for HCoV-OC43 siRNA-scr vs HCoV-OC43 siRNA-QTRT1; NP levels: $p = 0.042$ for HCoV-OC43 siRNA-

scr vs HCoV-OC43 siRNA-QTRT1. **F** NSUN2 levels: $p = 0.043$ for mock siRNA-scr vs mock siRNA-NSUN2; $p = 0.071$ for HCoV-OC43 siRNA-scr vs HCoV-OC43 siRNA-NSUN2; NP levels: $p = 0.91$ for HCoV-OC43 siRNA-scr vs HCoV-OC43 siRNA-NSUN2. Data are presented as mean ± s.d. Individual data points are shown. **G** Plasmid-mediated overexpression (OVE) of ADAT2 (I pathway) and ALKBH1 (hm⁵C/f⁵C pathway) in HCoV-OC43 (MOI 0.1) infected A549 cells. HCoV-OC43 NP levels was measured by western blot at 48 hpi. Empty vector transfection (pcDNA3) was used as a control. Vinculin was used as loading control. **H** Quantification of viral NP protein levels from independent biological replicates in plasmid-transfected cells relative to empty vector–transfected controls. Statistical significance was assessed using a two-sided, nonparametric Mann–Whitney U test (unpaired; $n = 4$). Exact p-values are $p < 0.0001$ for HCoV-OC43–empty vs HCoV-OC43–ADAT2 overexpression (OVE) and $p < 0.0001$ for HCoV-OC43–empty vs HCoV-OC43–ALKBH1 overexpression (OVE). Data are presented as mean ± s.d. Individual data points are shown.

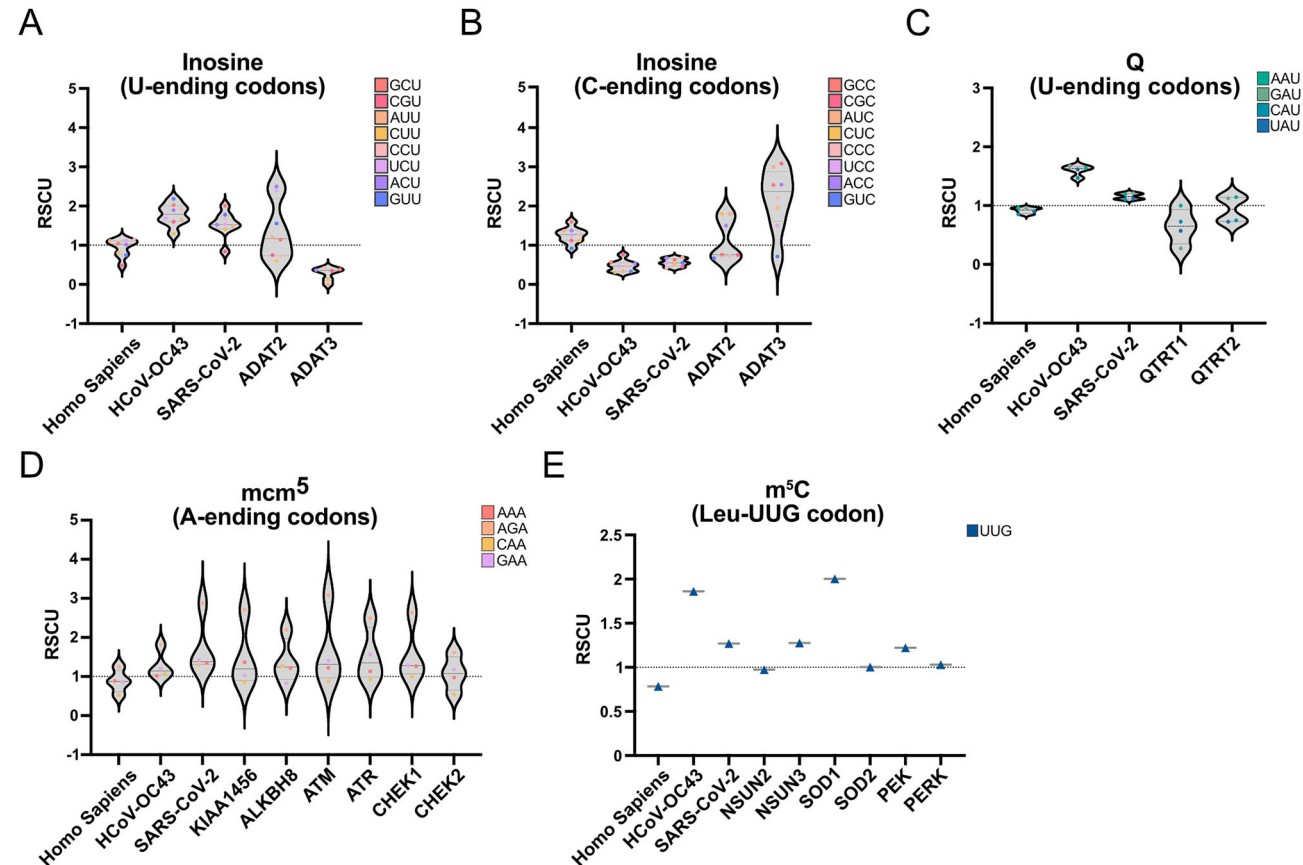

**Fig. 5 | RSCU analysis of tRNA modifying enzymes.** RSCU analysis of CDS of tRNA-modifying enzymes compared to HCoV-OC43 and SARS-CoV-2. **A** RSCU analysis of U-ending codons whose corresponding tRNAs can be modified by I in coronaviruses, along with the inosine writer enzymes ADAT2 and ADAT3. **B** RSCU analysis of C-ending codons that are favored by I modification. **C** RSCU analysis of U-ending codons that are favored by Q modification in coronaviruses, along with the Q writer enzymes QTRT1 and QTRT2. **D** RSCU analysis of A-ending codons that are favored by mcm5U/mcm5s2U modification in coronaviruses, along with the mcm5U writer enzymes KIAA1456 and ALKHB8, as well as DNA-damage response proteins. **E** RSCU analysis of the Leu-UUG codon in coronaviruses, along with oxidative stress response proteins and NSUN2 and NSUN3 writers enzymes.

Finally, to determine whether the expression of tRNA-modifying enzymes occurs at the transcriptional or translational level during infection, we analyzed their mRNA and protein abundance. qPCR analyses revealed that the mRNA levels were decreased (Supplementary Fig. 6A–F), whereas western blot analyses showed a concomitant increase in the protein levels (Fig. 3G–J and Supplementary Fig. 6G). These results suggest that expression of tRNA-modifying enzymes is primarily regulated at the translational level rather than transcriptionally during infection. Overall, our findings reveal a link between coronavirus codon usage, the observed alterations in tRNA modifications and in the corresponding tRNA-modifying enzymes, and viral protein expression.

**Coronaviruses reprogram modifications in pre-existing tRNAs without altering their abundance**

To determine whether coronavirus-induced alterations in tRNA modifications arise from changes in tRNA abundance or from modification of pre-existing molecules, and to identify possible alterations in the abundance of specific tRNA species, we employed mim-tRNAseq[45,46], a high-resolution method capable of quantifying tRNA abundance and modification status. tRNAs isolated from the same samples used for LC-MS analyses were processed for mim-tRNAseq. Quality control metrics, including alignment statistics and principal component analysis (PCA), confirmed high-quality sequencing data (Supplementary Figs. 7 and 8). For SARS-CoV-2, PCA revealed separate clusters for infected and non-infected samples, indicating distinct tRNAome alterations (Supplementary Fig. 7C, D). However, for HCoV-OC43, PCA

did not show clear differences between infected and non-infected samples (Supplementary Fig. 8C, D). In SARS-CoV-2-infected cells, of the 398 cytosolic tRNA species quantitatively resolved by mim-tRNA-seq, 24 displayed differential expression, with changes of less than one order of magnitude between mock-infected and SARS-CoV-2-infected samples (Fig. 6A, C). Importantly, none of these transcripts corresponded to the tRNAs affected by I, mcm5, Q, or f5C. When grouping them by the same anticodon (isoacceptors), three were upregulated (tRNA$^{Thr}_{TGT}$, tRNA$^{Cys}_{GCA}$, and tRNA$^{Lys}_{CTT}$) and seven downregulated (tRNA$^{Ser}_{GCT}$, tRNA$^{Ser}_{AGA}$, tRNA$^{Ser}_{TGA}$, tRNA$^{Ser}_{CGA}$, tRNA$^{Leu}_{CAA}$, tRNA$^{Leu}_{TAA}$, and tRNA$^{Leu}_{CAG}$) (Fig. 6B, D). In contrast, we found no significant changes in tRNA abundance in HCoV-OC43-infected MRC5 cells (Fig. 6E, F). When this occurs, PCA analysis typically cluster by the similarity of barcodes[46] (Supplementary Fig. 8C, D).

The observed changes in tRNA abundance in SARS-CoV-2 infected cells uncovered interesting features. The upregulation of tRNA$^{Thr}_{TGT}$ correlates with the high abundance of its cognate codon (Thr-ACA) in the SARS-CoV-2 genome, reflecting viral translational demands. Notably, this tRNA has been reported to be upregulated as part of the stress response[47]. In addition, the Lys-AAG codon, which corresponds to the upregulated tRNA$^{Lys}_{CTT}$, is also enriched in proteins associated with stress response, particularly short-term heat stress[48]. While the Lys-AAG codon is present at similar levels in both HCoV-OC43 and humans, its frequency is reduced in SARS-CoV-2. The upregulation of tRNA$^{Cys}_{GCA}$ is not linked to enriched viral codons, however, cysteine proteases are critical for SARS-CoV-2 infection[49]. Regarding the downregulated tRNAs, all belong to the type II tRNAs, which are

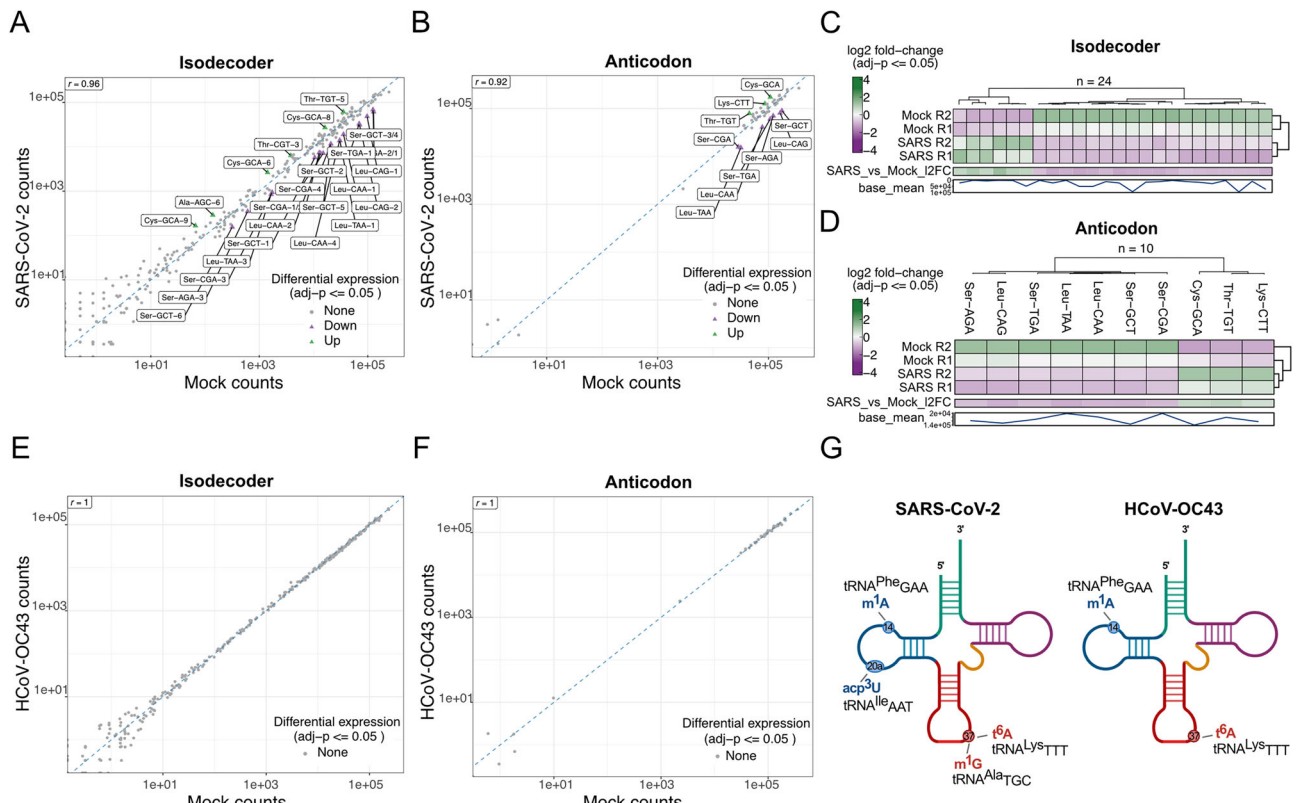

**Fig. 6 | SARS-CoV-2 infection triggers a decrease of type II tRNA transcripts.**
**A** Differential expression analysis of unique tRNA transcripts in SARS-CoV-2–infected Calu3 cells at 32 hpi relative to mock-infected cells. Axes represent log-transformed normalized read counts obtained using DESeq2. Significantly downregulated and upregulated tRNAs in infected cells are indicated by closed purple and green triangles, respectively, based on an FDR-adjusted one-sided Wald test (FDR ≤ 0.01; $n = 2$). **B** Differential expression analysis as in (**A**), performed on counts aggregated at the tRNA anticodon family level. **C**, **D** Summary of differential expression results for SARS-CoV-2–infected Calu3 cells relative to mock-infected controls, corresponding to panels (**A**) and (**B**). Bar plots show log2 fold changes, and numbers indicate the total counts of significantly upregulated (green) and downregulated (purple) tRNAs ($n = 2$). **E** Differential expression analysis of unique tRNA transcripts in HCoV-OC43–infected MRC5 cells at 48 hpi relative to mock-infected cells. Axes represent log-transformed normalized read counts from DESeq2, with significantly downregulated and upregulated tRNAs indicated by closed purple and green triangles, respectively, based on an FDR-adjusted one-sided Wald test (FDR ≤ 0.01; $n = 3$). **F** Differential expression analysis as in (**E**), performed on counts aggregated at the tRNA anticodon family level. **G** Graphical representation of tRNA modifications showing significant changes in SARS-CoV-2–infected Calu3 cells (MOI 3, 32 hpi; $n = 2$) relative to mock-infected controls and in HCoV-OC43–infected MRC5 cells (MOI 0.1, 48 hpi; $n = 3$) relative to mock-infected controls, as determined by mim-tRNAseq. Panel (**G**) was created in Biorender. Diez, J. (2026) https://BioRender.com/1xtg83d.

substrates for Schlafen 11 (SLFN11) and Schlafen 12 (SLFN12) endonucleases[50]. These enzymes are induced by the interferon (IFN) response as part of the innate antiviral defense or in response to DNA damage[51-54]. Consistent with the downregulation of type II tRNAs in SARS-CoV-2 infection, but not in HCoV-OC43 infection, SARS-CoV-2 infection triggered an IFN response, as indicated by the expression of IFN-stimulated genes and the strong upregulation of SLFN11 and SLFN12 (Supplementary Fig. 9A–C). In contrast, HCoV-OC43 did not induce a proper IFN response and only led to a slight activation of IFNB1 at 72 hpi, likely due to the DNA damage response (Supplementary Fig. 9B–D). These differences are linked to the cell lines used for SARS-CoV-2 and HCoV-OC43 experiments.

In addition to quantifying tRNA abundance, mim-tRNAseq can detect some specific modifications at defined tRNA positions[45,46]. In SARS-CoV-2 infected cells, four tRNA modifications were notably downregulated at position 14 in tRNA$^{Phe}_{GAA}$, position 20a in tRNA$^{Ile}_{AAT}$, and position 37 in tRNA$^{Ala}_{TGC}$ and tRNA$^{Lys}_{TTT}$ (Fig. 6G). The one at position 14 in tRNA$^{Phe}_{GAA}$ likely corresponds to m$^1$A, while position 20 in tRNA$^{Ile}_{AAT}$) is typically modified to acp$^3$U, enhancing tRNA stability and ribosomal interactions[9,55]. At position 37, modifications include m$^1$G in tRNA$^{Ala}_{TGC}$, which stabilizes codon-anticodon interactions, and t$^6$A in tRNA$^{Lys}_{TTT}$, essential for efficient codon recognition[9,55]. Similar patterns were observed for m$^1$A in tRNA$^{Phe}_{GAA}$ and t$^6$A in HCoV-OC43-infected MRC5 cells, with decreased m$^1$A in tRNA$^{Phe}_{GAA}$ and t$^6$A in tRNA$^{Lys}_{TTT}$ (Fig. 6G and Supplementary Fig. 10). These were also downregulated in LC-MS/MS analysis in HCoV-OC43 infected cells (Supplementary Fig. 4), indicating that both viruses reduce tRNA modifications at positions that potentially affect tRNA stability and translation fidelity.

Collectively, the mim-tRNAseq analysis shows that SARS-CoV-2, but not HCoV-OC43, alters tRNA abundance in a manner consistent with viral translational requirements and IFN activation mediated by SLFN11 and SLFN12. Importantly, the tRNAs exhibiting altered expression levels are distinct from those displaying modification changes, indicating that abundance and modification are regulated through separate mechanisms.

## Discussion
Our study reveals that coronavirus infection triggers a profound alteration of the tRNA modification landscape that aligns with the viral codon usage. Beyond coronaviruses, many RNA viruses have A/U-rich genomes, a bias suggested to arise from multiple selective pressures. For example, A/U-rich sequences may enhance RNA structural flexibility, thereby facilitating transitions between different stages of the viral life cycle[56,57]. Moreover, the underrepresentation of CpG dinucleotides likely reflects an evolutionary strategy to evade immune

detection[58]. Adding to these selective pressures, G-to-A hypermutations introduced by the error-prone reverse transcriptase in viruses such as HIV also contribute to A/U enrichment[59,60]. Since these factors are interconnected, it is difficult to dissect their individual contributions. Thus, despite extensive research, the selective forces shaping viral codon usage remain largely unresolved. Our findings support a positive selection pressure for the enrichment in coronavirus genomes of specific A- and U-ending codons that, although typically considered suboptimal, become optimal in the context of the modified tRNA landscape during viral infection.

Bioinformatic analyses determined that decoding of A- and U-ending codons highly enriched in coronavirus genomes depend on three key tRNA modifications: I, Q and $mcm^5U/mcm^5s^2U$. Using SARS-CoV-2 and HCoV-OC43 as models of highly and mildly pathogenic coronaviruses, respectively, we demonstrate that coronavirus infection induces extensive reprogramming of these tRNA modifications. This reprogramming favors the decoding of A- and U-ending codons over the G- and C-ending ones, enriched in highly expressed host transcripts, as summarized in Supplementary Fig. 11. Coronaviruses are unique in their exceptionally high percentage of U-ending codons. Indeed, all 16 U-ending codons occurred at higher frequencies compared to the host cell. Our comparative analyses across *Alphacoronavirus* and *Betacoronavirus* genera revealed that decoding of approximately 75% of these codons depends on the tRNA modifications I and Q. Decoding of the most highly enriched U-ending codons (e.g., Pro-CCU, Arg-CGU, Ala-GCU, Ser-UCU) is restricted by I modification (Fig. 1A, B). Interestingly, I facilitates decoding of A-ending codons also present in coronaviruses, suggesting that a reduction rather than a complete suppression, as observed in SARS-CoV-2 and HCoV-OC43 infections, would be required to support efficient translation of both U- and A-ending codons (Fig. 1). In turn, Q modification enhances the decoding of Asn-AAU, Asp-GAU, His-CAU, and Tyr-UAU. Notably, the four Q-favored codons are more enriched in mildly pathogenic coronaviruses than in highly pathogenic ones (Fig. 1). Consistently, only HCoV-OC43 infection, but not SARS-CoV-2, was found to upregulate Q (Fig. 3F), suggesting a potential link between viral pathogenicity and codon enrichment. Finally, the decoding mechanism of the highly enriched Gly-GGU codon remains unresolved. As no cognate tRNA is known for this codon, the involvement of yet-unidentified tRNA modification at the wobble position would be expected.

The decoding of four A-ending codons (Arg-AGA, Gln-CAA, Lys-AAA and Glu-GAA), highly enriched in coronavirus genomes and in host mRNAs encoding DNA damage response proteins[25] was facilitated by $mcm^5U/mcm^5s^2U$ modifications. Upregulation of $mcm^5U$ was observed in both cell culture and in hamsters infected with the SARS-CoV-2 ancestral strain D641G, which has lower infectivity compared to the SARS-CoV-2 strain used in our in vitro experiments. This upregulation was higher in samples with the highest viral load, suggesting that infection efficiency is a critical factor influencing the levels of tRNA modifications (Fig. 3E). Taken together, these findings suggest that infection efficiency influences tRNA modification levels, which might explain why the other tRNA modifications were not altered in vivo. Further in vivo experiments with a more suitable animal model will be required to clarify this matter.

From the G-ending codons, only the Leu-UUG one was found to be highly enriched in the coronavirus genome and in host mRNAs preferentially translated under oxidative stress conditions[18]. Oxidative stress in yeast drives the methylation of C34 in tRNA Leu-CAA to $m^5C$, which enhances translation of Leu-UUG-enriched transcripts. In addition, further oxidation of $m^5C$ to $f^5C$ expands decoding capacity to Leu-UUA codons, which would decrease Leu-UUG decoding. Our LC-MS/MS data did not detect a global change in $m^5C$ levels, however, we cannot exclude the possibility of site-specific alterations at position C34 in $tRNA^{Leu}_{CAA}$, as $m^5C$ can occur at multiple positions across

various tRNA and our assay measures total tRNA modification levels[61]. Moreover, the mim-tRNAseq technique cannot distinguish this modification in specific tRNAs. However, $f^5C$ levels were found to be reduced in both SARS-CoV-2- and HCoV-OC43-infected cells, which would favor decoding of the Leu-UUG codon (Fig. 3). Importantly, $f^5C$ also appears at the wobble position of mitochondrial tRNA-iMet, where it ensures correct decoding of AUA as methionine, thereby supporting efficient mitochondrial protein synthesis, oxidative phosphorylation, and energy metabolism. Additional studies will be required to determine whether the observed decrease in $f^5C$ affects cytoplasmic $tRNA^{Leu}_{CAA}$, mitochondrial tRNA-iMet, or both. In addition, other modifications unique to HCoV-OC43 infection conditions were altered. These occur at tRNA positions outside the wobble site and are involved in processes such as tRNA folding and stability, but remain to be assigned to specific tRNAs.

The differences in tRNA modification profiles observed between SARS-CoV-2 and HCoV-OC43 infections may stem from the distinct cell lines used, the virus themselves, or both. SARS-CoV-2 infections were conducted in the lung adenocarcinoma Calu3 cell line, whereas HCoV-OC43 infections were conducted in the lung fibroblast MRC5 cell line, both well-established and recommended models for their study.

While I and the corresponding tRNA modifying enzymes have been described to exhibit remarkable stability across various conditions[62], Q, along with $mcm^5U/mcm^5s^2U$ and $m^5C/f^5C$ modifications are dynamically regulated under stress conditions in a manner consistent with the alterations observed under coronavirus infection (Fig. 3). This indicates that under coronavirus infection conditions tRNA modifications and their associated enzymes can be modulated via two distinct mechanisms: one that is coronavirus-specific, affecting I, and another that is stress-related, affecting Q, $m^5C/f^5C$, and $mcm^5U/mcm^5s^2U$ modifications (Supplementary Fig. 11B). The described dynamic changes affecting Q, $m^5C/f^5C$, and $mcm^5U/mcm^5s^2U$ under stress conditions facilitate the selective expression of stress response proteins whose mRNAs, as coronavirus genomes, are enriched in codons favored by these modifications. This observation supports that coronaviruses have adapted their codon composition to optimize translation under stress response conditions triggered by infection. Under oxidative stress conditions, levels of Q and $m^5C/f^5C$ modification typically increase, whereas upregulation of $mcm^5U/mcm^5s^2U$ is primarily linked to DNA damage stress[23,25,63]. Our data show that infection of SARS-CoV-2 in Calu3 cells and of HCoV-OC43 in MRC5 cells triggers ATM phosphorylation, a hallmark of DDR activation, alongside increased levels of p-p38 MAPK, a central regulator of oxidative stress signaling (Fig. 2).

Mechanistically, the combination of LC-MS/MS and mim-tRNAseq revealed that alterations in the tRNA epitranscriptome are primarily driven by modifications of pre-existing tRNAs rather than by changes in tRNA abundance. This strategy enables coronaviruses to rapidly adapt the tRNA pool to their translational demands. Moreover, the coding sequences of the enzymes responsible for I and $mcm^5U/mcm^5s^2U$ modifications are themselves enriched with the very codons whose translation they enhance, creating a positive feedback loop that may amplify the effect of the coronavirus-induced tRNA modification changes. Importantly, manipulating the expression of ADAT2 (I-pathway), QTRT1 (Q-pathway), and ELP3 ($mcm^5U/mcm^5s^2U$-pathway) tRNA modifying enzymes altered coronavirus protein levels (Fig. 4), providing a functional causal link between the observed tRNA epitranscriptome alterations and viral protein synthesis. In contrast, siRNA-mediated NSUN2 knockdown ($m^5C$ pathway; cytoplasm) did not affect viral protein levels, suggesting that cytoplasmic $m^5C$ pathway does not play a relevant role in supporting viral protein production, while overexpression of ALKBH1 ($f^5C$-pathway; cytoplasm and mitochondria) significantly affected viral protein expression, suggesting a role of the mitochondrial NSUN3–ALKBH1 pathway rather than the cytosolic NSUN2-dependent $m^5C$ modification (Fig. 4).

Overall, because many viruses induce oxidative stress, trigger DNA damage responses, and are enriched in A/U ending codons, we propose that viral adaptation to the codon optimality under stress conditions represents a shared viral strategy to enhance viral protein production. In support of this notion, (+)RNA viruses such as chikungunya and dengue viruses, which are enriched in mcm⁵U/mcm⁵s²U-dependent codons, elicit DNA damage responses that enhance mcm⁵U modification. These findings highlight the potential of targeting tRNA-modifying enzymes as a promising strategy for developing broad-spectrum antivirals. Moreover, if such compounds achieve clinical approval, their use could be guided by codon usage analysis, even for newly emerging coronaviruses whose biological properties remain unknown. In addition, codon analyses may help elucidate the timescale of viral adaptation to humans, as some codons associated with tRNA modifications exhibited are more enriched in mild coronavirus strains that have circulated in humans for longer periods compared to those that have emerged more recently (e.g., codons Val-GUU, Pro-CCU, Leu-UUG, Asn-AAU). This understanding might ultimately enable predictions of pathogenicity based on codon adaptation patterns.

## Methods

### Cell lines
MRC5 (kindly provided by Wolfram Brune), A549 (kindly provided by Ana Janic), and Calu3 (kindly provided by Alfredo Castello) cells were grown at 37 °C and 5% CO2. MRC5 and A549 cells were cultivated in Dulbecco's modified Eagle's medium (DMEM, ThermoFisher, 41966-029) supplemented with 10 % fetal bovine serum (FBS, Sigma F7524), 1 % non-essential amino acids (NEAA, Gibco 11140-035) and 1% Penicillin/Streptomycin 10.000 U/ml (Pen/Strep, ThermoFischer 15140163). Calu3 cells were cultivated in DMEM supplemented with 20 % FBS and 1% Pen/Strep.

### Viruses and infection conditions
BHK21 (kindly provided by Andres Merits) and VERO-E6 (ATCC CRL-1587) cells were used to produce viral stocks. BHK21 cells were grown at 33 °C and 5% CO2 and cultivated in Glasgow Minimum essential medium (GMEM, Lonza BE12- 739F), supplemented with 10% FBS, 10% triptose phosphate buffer and 1 M Hepes pH 7.2 (Gibco 15630080). VERO-E6 cells were grown at 37 °C and 5% CO2 and cultivated in DMEM supplemented with 10 % FBS, 1% NEAA and 1% Pen/Strep. Stocks of the SARS-CoV-2 strain hCoV-19/Spain/VH000001133/2020 (EPI_ISL_418860) (kindly provided by Miguel Chillon Rodriguez) were generated in VERO-E6 cells and titrated by qPCR assay. HCoV-OC43 was purchased at ATCC (VR-1558TM) and was propagated in BHK21 cell lines and titrated in the same cell line by TCID50. Titers were obtained four days post-infection and calculated using the Spearman-Kärber formula. SARS-CoV-2 infections were carried out in Calu3 cells. Briefly, cells were seeded and the day after were washed twice with phosphate-buffered saline (PBS, Cultek SH30028.02) and incubated with SARS-CoV-2 at an MOI of 3 for 1 h in DMEM containing 1% FBS. Plates were shaken every 10 minutes (min). After incubation, the virus was removed, and pre-warmed DMEM + 10% FBS was added. HCoV-OC43 infection was carried out in MRC5 cells and A549 cells. Briefly, cells were seeded and the day after were washed twice with PBS and incubated with HCoV-OC43 at an MOI of 0.1 for 2 h in DMEM containing 1 % FBS. Infection was carried out at 37 °C for MRC5 cells and 33 °C for A549 cells. Cells were harvested and analyzed at the indicated time points. Golden Syrian Hamsters were intranasally infected with the ancestral SARS-CoV-2 carrying the D614G mutation (viral load 10⁴) in spike (n = 6) as described previously, and lungs were collected 2 days post-infection. Uninfected hamsters were used as controls (n = 6)[64]. Tissues were harvested and immediately homogenized in TRIzol (Thermo Fisher Scientific 15596018). Phase separation was achieved by adding 200 μL of chloroform per 1 mL of TRIzol, shaking vigorously for 15 seconds, and incubating at RT for 2 min. Samples were centrifuged

at 12,000 × g for 15 min at 4 °C, and the aqueous phase was transferred to new tubes. RNA was precipitated by adding 500 μL of isopropanol per 1 mL of TRIzol, incubating for 10 min at RT, followed by centrifugation at 12,000 × g for 10 min at 4 °C. The RNA pellet was washed with 75% RNase-free ethanol (1 mL per 1 mL of TRIzol), centrifuged at 7500 × g for 5 min at 4 °C, and air-dried for 5–10 min. Pellets were resuspended in RNase-free water and incubated at 55–60 °C for 10 min. RNA was either stored at − 80 °C or immediately processed for DNase treatment with TURBO DNase (Thermo Fisher Scientifi AM1907c). Reactions were incubated at 37 °C for 1 h, followed by DNase Inactivation for 5 min at 37 °C. Samples were centrifuged at 10.500 × g for 1.5 min, and the supernatant containing RNA was transferred to new tubes. RNA concentration was measured using a Nanodrop spectrophotometer.

### Immunodetection
For immunoblot analysis of cell lysates, cells were lysed at the indicated time points in boiling 2x SDS-PAGE sample buffer (125 mM Tris-HCl, pH 6.8, 4% SDS, 20% glycerol, 10% β-mercaptoethanol) and incubated 10 min at 95 °C. Samples were separated on denaturing SDS-PAGE gels and transferred electrophoretically onto nitrocellulose (trans-blot turbo, BioRad 12023835) by turbo-blotting (trans-blot turbo transfer system, BioRad 1704150EDU). Antibodies recognizing the following epitopes and proteins were used: SARS-CoV-2 NSP8 (GeneTex GTX632696, 1:1000); KIAA1456 (Thermo Fisher PA4-70320, 1:500); ALKBH8 (AV41106Sigma-Aldrich AV41106, 1:1000); ADAT2 (mybiosurce mbs2621771, 1:1000); ALKBH1 (Abcam ab126596, 1:1000); NSUN2 (Proteintech 20854-1-AP, 1:5000); QTRT1 (D7) (Santa Cruz Biotechnology sc398918, 1:1000); Vinculin (Sigma-Aldrich V9131, 1:1000); GAPDH (Proteintech 60004-1-IG, 1:10000); Phospho-histone H2AX (Ser139) (Cell Signaling 9718S, 1:1000); ATM (D2E2) (Cell Signaling 2873S, 1:1000); Phospho-ATM (S1981) (Cell Signaling 13050S, 1:1000); ATR (Cell Signaling 2790S, 1:1000); Phospho-ATR (S428) (Cell Signaling 2853S, 1:1000); Phospho-p38 MAPK (Thr180/Tyr182) (Cell Signaling 9215T, 1:1000); SLFN11 (E4) (Santa Cruz Biotechnology sc374339, 1:1000); SLFN12 (Abmart X-C9J4K7-N, 1:1000); HCoV-OC43 Nucleocapsid Antibody (NP) (SinoBiological -abyntech 40643-T62, 1:20000); ELP3 (D5H12) (Cell Signaling 5728, 1:1000). For detection, HRP-conjugated secondary antibodies (goat anti-rabbit IgG-HRP or goat anti-mouse IgG-HRP, Jackson ImmunoResearch GENA934 and GENA931, 1:5000) were used in combination with chemiluminescent substrate (Cytiva 10600119) for standard immunoblotting. Alternatively, fluorescent secondary antibodies (StarBright Blue 700, Bio-Rad 12004161, 1:2500) were employed, and signals were visualized using a ChemiDoc MP Imaging System (Bio-Rad). Uncropped and unprocessed western blot scans are provided in the Supplementary Information File.

### RNA extraction and analysis
Extracellular RNA was extracted from 400 μL supernatant using ZR small RNA PAGE Recovery kit (Zymo Research R1070) following the manufacturer's instructions. Intracellular RNA was extracted from cells by phenol-chloropropane extraction, after cell lysis with a buffer containing 10 mM Tris-HCl pH 7.5, 10 mM MgCl2, 100 mM NaCl, 10% Triton, and 5% DTT 0.1 M. Lysates were incubated 10 min at 60 °C with proteinase K (NEB), phenol-chloropropane extracted and precipitated with ethanol. The pellet was resuspended in RNAse-free water and treated with 0.1 μl TURBO DNase (Thermo Fisher Scientifi AM1907c) for 30 min at 37 °C. Viral RNA was analyzed by qPCR using qScript XLT One-Step RT-qPCR ToughMix (Quanta BioSciences 733-2232) using 18 ng of total RNA. RNA were reverse transcribed and amplified using gene- specific primers and TaqMan probes for all analyzed genes. The following primers for HCoV-OC43 N-protein were used: Forward *TCAGCGTGGTCATAAGAATGG*; Reverse *GTGTCTTCAGTATAGGGCTCATC*; Probe 56-FAM/*ATAAGTGTTG-CAGTGCCCAAAAGCC*/36-TAMSp. The following primers for SARS-CoV-

2 N-protein were used: Forward *GACCCCAAAATCAGCGAAAT*; Reverse *TCTGGTTACTGCCAGTTGAATCTG*; Probe 56-FAM/*ACCCCGCATTACGTTTGGTGGACC*/BHQ1.

For cellular genes, 500 ng of total RNA was initially retro-transcribed using the SuperScript™ III One-Step RT-PCR System with Platinum™ Taq DNA Polymerase (ThermoFisher 12574026) with random primers (ThermoFisher N8080127). Subsequently, quantitative real-time polymerase chain reaction (RT-qPCR) was performed using the PowerTrack™ SYBR Green Kit (ThermoFisher A46012). The following primers were used: IRF3 Forward *TCTGCCCTCAACCGCAAA-GAAG*; IRF3 Reverse *TACTGCCTCCACCATTGGTGTC*; MX1 Forward *GGCTGTTTACCAGACTCCGACA*; MX1 Reverse *CACAAAGCCTGG-CAGCTCTCTA*; ISG15 Forward *CTCTGAGCATCCTGGTGAGGAAl*; ISG15 Reverse *AAGGTCAGCCAGAACAGGTCGT*; OAS1 Forward *AGGAAAGGTGCTTCCGAGGTAG*; OAS1 Reverse *GGACTGAGGAAGA-CAACCAGGT*; IFNA1 Forward *AGAAGGCTCCAGCCATCTCTGT*; IFNA1 Reverse *TGCTGGTAGAGTTCGGTGCAGA*; IFNB1 Forward CTTGGATTCCTACAAAGAAGCAGC; IFNB1 Reverse *TCCTCCTTCTGGAACTGCTGCA*; GAPDH Forward *TGTCAGTGGTG-GACCTGAC;* GAPDH Reverse *GTGGTCGTTGAGGGCAATG.* qPCR were executed in the QuantStudio™ 3 real-time PCR system from (ThermoFisher A28567). The relative changes in gene expression were determined using the 2-ΔCt (cycle threshold) quantification method, normalized to the expression levels of GAPDH. A significance test was conducted using a Student's *t* test with GraphPad Prism v10.

### siRNA-mediated knockdown and plasmid overexpression

A549 cells were transfected with 5 pmol siRNA - SMARTpool toward ELP3 (Cultek L-015940-01-0005) and QTRT1 (Cultek L-007081-01-0005) using Lipofectamine RNAiMAX (ThermoFisher 13778030) for 24 h, followed by infection with HCoV-OC43 at a MOI of 0.1. At 2 hpi, the inoculum was removed, and cells were subjected to a second round of siRNA transfection. Cells, supernatants, and protein lysates were harvested at 48 hpi for downstream analyses. A non-targeting siRNA pool (scr) was used as a control (Cultek D-001810-10-05). For overexpression, cells were transfected with 500 ng plasmid DNA encoding ALKBH1-myc-ddk (Origene SKURC206529), ADAT2-myc-ddk (Origene SKURC236014), or an empty pcDNA vector for 24 h, followed by infection with HCoV-OC43 (MOI 0.1). At 2 hpi, the input virus was removed, and cells were re-transfected with the respective plasmid. FuGENE HD (FuGENE Labclinics HD-1000) was used as a transfection reagent in a ratio of 1:4. Protein, RNA, and supernatant samples were collected at 48 hpi for analysis.

### DCFH-DA assay

Intracellular ROS levels were measured using the fluorescent probe 2′,7′-dichlorodihydrofluorescein diacetate (DCFH-DA Sigma-Aldrich D6883). A549 cells were infected with HCoV-OC43 at an MOI of 0.1 and incubated for 48 h at 33 °C. Cells were washed twice with PBS and incubated with 10 μM DCFH-DA in serum- and phenol-free medium (DMEM, high glucose, HEPES, no phenol red, Gibco 11520556) for 30 min at 37 °C in the dark. After staining, cells were washed, detached with phenol-free trypsin, and immediately analyzed by flow cytometry using a BigFoot Spectral Cell Sorter (ThermoFisher Scientific). Fluorescence was collected in the FITC channel (488), and ROS levels were quantified by calculating the mean fluorescence intensity (MFI, CT mean) of the cell population using FlowJo software (BD Biosciences). For positive control treatment, cells were incubated with 1 mM $H_2O_2$ for 30 min prior to DCFH-DA staining.

### ATM inhibitor treatment

A549 cells were pretreated with the ATM inhibitor CP-466722 (5 μM, MedChemExpress HY-11002) for 1 h prior to infection. Cells were then infected with HCoV-OC43 at an MOI of 0.1 for 2 h at 33 °C. Following removal of the inoculum, cells were replenished with fresh medium

containing CP-466722 (10 μM), which was maintained for the duration of the experiment. At 48 hpi, cells were harvested, and protein lysates were prepared for Western blot analysis.

### tRNA extraction and analysis by LC-MS/MS

The tRNA fraction was isolated from 5 up to 10 μg of total RNA. tRNAs were separated using 10 or 15% Novex TBE-Urea gels (Novex, ThermoFisher Scientific EC6875BOX or EC6885BOX). The RNA was mixed with the same amount of Novex TBE-UREA sample Prep Buffer (2x) (Novex LC6876), loaded into the gels and run at 180 V for 60 min in 1X Novex TBE Running Buffer (Invitrogen LC6675). Gels were stained with 1:5000 SybrGold (ThermoFisher Scientific S11494), and the tRNA bands were excised. tRNA was then extracted using the ZR small RNA PAGE Recovery kit (Zymo Research R1070) following the manufacturer's instructions. 150 ng of the obtained tRNA was digested with 1 μL Nucleoside Digestion Mix (New England Bio Labs M0649S).

### Chromatographic and mass spectrometric analysis

Samples (15 ng) were analyzed using an Orbitrap Eclipse Tribrid mass spectrometer (Thermo Fisher Scientific, San Jose, CA, USA) coupled to an EASY-nLC 1200 (Thermo Fisher Scientific (Proxeon), Odense, Denmark). Ribonucleosides were loaded directly onto the analytical column and were separated by reversed-phase chromatography using a 50 cm column with an inner diameter of 75 μm, packed with 2 μm C18 particles (Thermo Fisher Scientific, cat # ES903). Chromatographic gradients started at 93% buffer A and 3% buffer B with a flow rate of 250 nl/min for 5 min and gradually increased to 30% buffer B and 70% buffer A in 20 min. After each analysis, the column was washed for 10 min with 0% buffer A and 100% buffer B. Buffer A: 0.1% formic acid in water. Buffer B: 0.1% formic acid in 80% acetonitrile. The mass spectrometer was operated in positive ionization mode with nanospray voltage set to 2.4 kV and source temperature at 305 °C. For the Parallel Reaction Monitoring (PRM) method, the quadrupole isolation window was set to 1.4 m/z, and MSMS scans were collected over a mass range of m/z 50–300, with detection in the Orbitrap at a resolution of 60,000. MSMS fragmentation of defined mases at scheduled retention time (Supplementary Data 1) was performed using HCD at NCE 20 (except stated differently, Supplementary Data 1)[65], the auto gain control (AGC) was set to "Standard" and a maximum injection time of 118 ms was used. In each PRM cycle, one full MS scan at a resolution of 120,000 was acquired over a mass range of m/z 220-700 with detection in the Orbitrap mass analyzer. Auto gain control (AGC) was set to 1e5, and the maximum injection time was set to 50 ms. Serial dilutions were prepared using commercial pure ribonucleosides (0.005-150 pg, Carbosynth, Toronto Research Chemicals) in order to establish the linear range of quantification and the limit of detection of each compound. A mix of commercial ribonucleosides was injected before and after each batch of samples to asses instrument stability and to be used as an external standard to calibrate the retention time of each ribonucleoside (Supplementary Data 1).

### LC-MS/MS Data analysis

Acquired data were analyzed with the Skyline-daily software (v24.1.1.284), and extracted precursor areas of the ribonucleosides were used for quantification (Source Data File).

### Ribosome profiling data processing

We re-analyzed publicly available ribosome profiling and RNA-seq datasets of SARS-CoV-2 infection in Calu3 cells at 36 h post-infection (hpi) from NCBI GEO (accession number GSE157490)[43]. Adapters were trimmed using *cutadapt* (v4.2), retaining reads ≥ 50 nt for RNA-seq and 25–35 nt for Ribo-seq. Reads mapping to non-coding RNAs (tRNAs, rRNAs, and other ncRNAs) were removed by alignment with *bowtie2* (v2.4.5). For human transcripts, we used the GENCODE v48 annotation (GRCh38.p14, GCA_000001405.29) and restricted the analysis to the

MANE Select transcript isoforms, as defined by the Appris database (appris_data.principal_mane_match_G48.tsv). The SARS-CoV-2 reference genome and transcriptome were obtained from GenBank (NC_045512.2) and combined with the human annotation. Processed reads were aligned to this merged reference using *STAR* (v2.7.10b). RNA-seq counts were quantified with *featureCounts* (subread v1.5.1). Assignment of A- and P-sites and ribosome footprint counting were performed using *riboWaltz* (v2.0), considering footprints of length 29–33 nt and determining offsets with the "extremity = auto" parameter. Differential expression analyses were performed using *DESeq2* using two-sided Wald tests[66] for RNA-seq, and Ribo-seq counts independently. Translation efficiency was calculated using *RiboDiff*[67]. Genes were classified as translationally activated (TA), translationally repressed (TR), or not significantly changed (NS). Codon usage was computed using the uco function from the *seqinR* R package (v4.2-36). Occurrences of G-, C-, A-, and U-ending codons were determined for each transcript, and relative usage of codons recognized by inosine (I)-tRNAs, queuosine (Q)-tRNAs, and mcm$^5$U-tRNAs was calculated. For Q-family codons, the percentage of each individual NNC and NNU codon was calculated as its mean occurrence in the corresponding gene set divided by the total occurrence of all Q-family codons (Supplementary Data 2). For every sample, ribosome occupancy density of each gene was calculated by using riboWaltz in built function codon_usage_psite and normalized to the total counts of RPFs per gene[68]. Then, the average codon occupancy in SARS-infected conditions was divided by non-infected conditions for codons associated with tRNA modifications. Results were visualized using ggplot.

## mim-tRNAseq

The analysis of tRNA expression was performed using the mim-tRNAseq protocol as described[45,46]. Briefly, total RNA was extracted from Calu3 cells infected with SARS-CoV-2 and MRC5 cells infected with HCoV-OC43 using phenol-chloroform extraction, followed by precipitation with ethanol. RNA was deacylated using Tris-HCl (pH 9), followed by dephosphorylation with T4 PNK (NEB M0201S). Next, tRNA was purified on a 10% TBE-Urea gel (Novex, Thermo Fisher Scientific EC6875BOX), size-selected (60–100 nt), and eluted from the gel using the ZR Small RNA PAGE Recovery Kit (Zymo Research R1070) according to the manufacturer's instructions. Preadenylated barcoded 3′ adapter oligonucleotides (containing a 5′ phosphate and blocked with a 3′ dideoxycytidine) were ligated to the purified tRNA using T4 RNA ligase 2 truncated KQ (NEB M0373L), in a reaction containing 50% PEG-8000, T4 RNA ligase buffer (10X), and RNase inhibitor (SUPERase•In, Invitrogen AM2694) for 3 hours at 25 °C. The following barcoded 3′ adapters were used (sequences provided 5′ to 3′): *pGATATCGTCAAGATCGGAA-GAGCACACGTCTGAAddC*; pGATAGCTACAAGATCGGAAGAGCACACG TCTGAAdd*C*;pGATGCATACAAGATCGGAAGAGCACAC*GTCTGAAddC*; *pGATTCTAGCAAGATCGGAAGAGCACACGTCTGAAddC*. Reaction products with different barcoded 3′ adapters were pooled and purified with the Oligo Clean & Concentrator Kit (Zymo Research D4060) before size selection by gel (110–125 nt), as described above. A total of 100 ng of adapter-ligated tRNA was used for a primer-dependent reverse transcription reaction containing 125 nM RT primer (5′-*pRNAGATCGGAA-GAGAGCGTCGTGTAGGGAAAGAG/iSp18/GTGACTGGAGTTCAGACGTGT GTGCTC*-3′) previously annealed at 82 °C for 2 min 500 nM TGIRT (InGex) in FS 5X buffer (SuperScript III, Thermo Fisher 18080085), freshly prepared DTT (0.5 mM), RNase inhibitor, and dNTPs (25 mM), and incubated at 42 °C for 16 h. After reverse transcription, 1 µl of 5 M NaOH was added, and the RNA was hydrolyzed for 5 min at 95 °C. cDNA products were circularized using CircLigase ssDNA Ligase (Lucigen CL4111K) in a reaction containing CircLigase buffer (10X), ATP (1 mM), MnCl$_2$ (50 mM), and betaine (5 M) for 3 h at 60 °C. The circularized cDNA was used for library construction PCR using KAPA HiFi DNA Polymerase (Roche, KK2101) in KAPA HiFi GC buffer (5X) with a common forward primer(5′*AATGATACGGCGACCACCGAGATCTACA*

*CTCTTTCCCTACACGACGCT\*C-3′*) and specific indexed reverse primers with the reverse complement of an Illumina index sequence (sequences provided 5′ to 3′): *CAAGCAGAAGACGGCATACGAGATTGACCAGTGA*C TGGAGTTCAGACGTGT*G*; CAAGCAGAAGACGGCATACGAGATACAGT GGTGACTGGAGTTCAGACG*TGT\*G*; *CAAGCAGAAGACGGCATACGAGA TGCCAATGTGACTGGAGTTCAGACGTGT\*G*. Amplified libraries were purified using the DNA Clean & Concentrator Kit (Zymo Research D4013), pooled, and purified by size selection on an 8% non-denaturing polyacrylamide gel (150–230 bp). Libraries were quantified with real-time PCR and sequenced on an Illumina NextSeq 500 platform with a single-end run of 150 cycles. Read demultiplexing and adapter trimming were performed with Cutadapt (v3.7). Both read ends were quality-trimmed (-q 30,30), and reads shorter than 10 nt were discarded (-m 10). Indels were not allowed (--no-indels), and only trimmed reads were retained (--trimmed-only). An additional round of trimming was performed to remove two extra 5′ nucleotides introduced by cDNA circularization (-u 2). tRNA expression and modification analysis were processed using the mim-tRNAseq computational pipeline (v1.3.8) with the following parameters: --species Hsap --cluster-id 0.97 --threads 5 --min-cov 0.0005 --max-mismatches 0.1 --max-multi 4 --remap --remap-mismatches 0.075.

## Relative synonymous codon usage analysis (RSCU)

Viral coding sequences were retrieved from the NCBI GenBank. Only high-quality, annotated CDS were included in the analysis. Redundant sequences, pseudogenes, and incomplete coding regions were excluded to ensure accurate codon frequency calculations. For *Alphacoronaviruses* and *Betacoronaviruses*, coronavirus genes included ORF1ab, spike, envelope, membrane protein, and nucleocapsid genes, concatenating these sequences in the aforementioned order. Genes such as 3a/b, 4, and 7b were excluded from the study due to annotation inconsistencies. The human CDS genome was derived from a custom reference described[10]. Briefly, the hg38 human genome sequence was from the USCS Genome Browser and the most highly expressed protein-coding splice isoform in HEK293T RNA-seq data were selected using Appris principal isoforms annotation based on GENCODE v24. The sequence of each transcript was concatenated and RSCU was calculated via the "uco" function from the seqinR package[69]. RSCU values of less than 0.9 are considered underrepresented codons and RSCU values above 1.1 are deemed over represented codons. The R code utilized in this study is available from the corresponding author upon request.

## Large language models (LLMs) usage

During the preparation of this manuscript, ChatGPT was utilized to assist with grammatical corrections. All content was reviewed and approved by the authors to ensure accuracy and integrity.

## Ethics and Inclusion statement

Procedures involving animals were performed under UK Home Office License PP0271643 in accordance with the Animals (Scientific Procedures) Act 1986 and approved by the University of Glasgow Ethics Committee. All animal research adhered to ARRIVE guidelines[70].

## Reporting summary

Further information on research design is available in the Nature Portfolio Reporting Summary linked to this article.

# Data availability

The minimum dataset that supports the findings of this study is provided in this paper. The western blot scans generated in this study are included in the Supplementary Information. The data presented in graphs within the figures are provided in the Source Data File. The ribosome profiling analyses are provided in Supplementary Data 2. The mim-tRNA-seq sequencing data have been deposited in BioProject

under the accession number PRJNA1262447. The mass spectrometry raw data have been deposited in MetaboLights under the accession number MTBLS13738[71]. Source data are provided in this paper.

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

## Acknowledgements

This work was supported by the Spanish Ministry of Science, Innovation and Universities (PID2022-136939OB-I00 funded by MICIU/AEI/10.13039/501100011033 and PID2020-118602RB-I00), by the 2021 SGR 00176 and 2021 SGR 00529 grants from the Departament de Recerca i Universitats de la Generalitat de Catalunya and an institutional Unidad de Excelencia María de Maeztu CEX2024-001431-M, funded by MICIU/AEI/10.13039/501100011033. E.M. was the recipient of Marie Skłodowska-Curie Grant Agreement No 101065094. A.H.P. acknowledges support from the Life-Arc COVID-19 award and the MRC core award (MC_UU_00034/9). A.C. is funded by the European Research Council (ERC) Consolidator Grant 'vRNP-capture' N#101001634, and the MRC grant MC_UU_00034/2. The mass spectrometry analyses were performed in the CRG/UPF Proteomics Unit, which is part of the Spanish National Infrastructure for Omics Technologies (ICTS OmicsTech). The Flow Cytometry analyses were performed in the CRG/UPF Flow Cytometry Unit, which is part of the Spanish National Infrastructure for Omics Technologies (ICTS OmicsTech). We thank Andreas Meyerhans for fruitful discussions.

## Author contributions

E.M. designed the experiments, performed the experiments, wrote the manuscript, and managed the budget. M.P.-T., J.B.B., D.C.M., M.T.-P., G.P.-V., O.C.-P., G.R.M., and K.K. performed experiments. M.P-T and M.T.-P. also contributed to the scientific discussion. V.H., M.C., A.C., R.S., and A.H.P. provided materials; A.C., R.S., and A.H.P. also contributed to scientific discussion. J.D. designed experiments, contributed to scientific discussion, managed the budget, and wrote the manuscript.

## Competing interests

The authors declare no competing interests.
