## [Transparent Peer Review file · Nature Communications]

Coronaviruses reprogram the tRNA epitranscriptome to favor viral protein expression

Corresponding Author: Professor Juana Diez Anton

Version 0:

Reviewer comments:

Reviewer #1

(Remarks to the Author)

In their manuscript "Coronaviruses reprogram the tRNA epitranscriptome to favor viral protein expression." The authors demonstrate that coronavirus genomes are enriched with suboptimal A- and U-ending codons. They propose that this bias reflects an adaptation of viral codon usage to the host tRNA modification landscape under stress conditions induced by infection. The observations and underlying hypothesis are intriguing and may exemplify how the viral transcriptome has coevolved with the host tRNA modification machinery to optimize viral translation efficiency. However, the manuscript relies entirely on observational data and correlations, lacking direct cause-and-effect evidences and experimental validation.

This limitation significantly weakens the strength and overall impact of the study.

To validate their hypothesis, the authors should consider manipulating the host cell tRNA modification machinery, using host cells that either lack or overexpress the tRNA modification enzymes implicated in viral adaptation. One might expect reduced viral protein expression in knockout cells deficient in specific enzymes, such as Qtrt1/Qtrt2, as well as in cells overexpressing ADAT2. The same principle applies to NSun2 and ALKBH8.

In addition, I have the following major issues:

Mass spectrometry reveals variations in overall modifications upon infection but fails to demonstrate site-specific changes. While some observed trends align with the author's hypothesis, it remains unclear whether the changes are occurring on the expected tRNAs at position N34 and whether they result from an increase or decrease in modifications at these sites. The observed changes may be due to variations in tRNA expression, as suggested in Figure 5, which would inevitably affect the modification-to-tRNA ratio.

To validate conclusions regarding changes in tRNA modifications upon viral infection, a modification/tRNA-targeted approach is necessary.

Moreover, since mitochondrial tRNA modifications should not directly affect viral protein translation, their contribution, along with mit-tRNA expression, to the overall observed tRNA modification changes remains unclear.

Even if some modification derivatives are evaluated and discussed, others like Q/manQ/galQ (PMID: 37992713) are completely ignored, especially in their impact on the viral codon translation.

Minor

Page 4 row 136 the statement " When Q-modified, G34, which normally pairs with C or with near-cognate Q-decoded codons (such as Glu-GAA and Glu-GAG), can also base pair with U, disfavoring decoding of coronavirus codons" is in contradiction with Figure 1F. Please clarify.

Reviewer #2

(Remarks to the Author)

In this study, authors found that SARS-CoV-2 and HCoV-OC43 infection induces extensive reprogramming of inosine,

queuosine, mcm5U/mcm5s2U and m5C/f5C tRNA modifications. It is an interesting study that contributes to our understanding of host-virus interactions, particularly how viruses can exploit epitranscriptomic mechanisms to improve viral protein expression. However, there are several specific concerns and results interpretations that need to be addressed to support stated conclusions and strengthen the significance of these findings.

Major Comments

1. The authors used p-p38 to analyze oxidative stress. The activation of p38 MAPK is involved in various biological processes. Therefore, p-p38 is not an ideal biomarker of oxidative stress. The authors should provide ROS or MDA analysis data to confirm the oxidative stress.
2. The authors used MOI 3 for SARS-CoV-2 infection, MOI 0.1 for OC43 infection. Why use such a high MOI for SARS-CoV-2? Are similar tRNA changes observed at MOI 0.1? If yes, please provide the results from low MOI to higher MOI conditions. Were these changes only observed under high MOI conditions? If yes, why?
3. In figure 3, authors showed the up/downregulation of inosine, queuosine, mcm5U, and f5C in cells. However, only the increase of mcm5U was further confirmed in vivo. The other tRNA changes also need to be confirmed in vivo.
4. In figure 3, the OC43 infection induced more extensive tRNA modification changes than SARS-CoV-2. However, in figure 5, the host tRNA abundance only was observed in SARS-CoV-2 infection but not in OC43 infection. Is it reasonable for such a phenomenon to occur? Please clarify this point. And the title "Coronaviruses alter the host tRNA abundance" of this section should be changed.
5. In Figure 6, it is unclear whether the same MOIs were used as in previous experiments—MOI 3 for SARS-CoV-2 and MOI 0.1 for OC43. If this is the case, the observed differences in SLFN11/12 expression and interferon-related genes expression between the two viruses may be influenced. The authors should clarify the MOIs used in this figure and discuss how differences in viral dose may impact the interpretation of host response data.

Minor Comments

1. The title claims that coronaviruses reprogram the tRNA epitranscriptome to favor viral protein expression. Therefore, the increase of viral protein levels is a critical phenomenon. The authors should show viral protein expression levels in main figures not just in supplementary figures. Extended figure 2 showed only SARS-CoV-2 protein levels. The authors should add OC43 viral protein expression data.
2. The authors propose that SLFN11/12 upregulation leads to a decrease in type 2 tRNA abundance under SARS-CoV-2 infection. However, the observed timing is inconsistent: the reduction in type 2 tRNAs occurs at 32 hours post-infection, whereas SLFN11/12 expression increases later, at 48 hours. This temporal discrepancy raises questions about whether SLFN11/12 induction is the cause of the tRNA reduction or a downstream consequence. Clarification of this timeline, or additional data supporting a delayed effect of SLFN11/12, would strengthen the authors' conclusion.
3. Add a graphical abstract or summary figure may help readers grasp the key findings.

Reviewer #3

(Remarks to the Author)

Coronavirus genomes are enriched in suboptimal codons, which potentially reduce the rate of protein synthesis. However, coronavirus proteins can be efficiently translated despite the codon usage conflict. To tackle this inconsistency, the authors uncovered that coronavirus infection changes tRNA-modifying enzyme expression and tRNA epi-transcriptomics to modulate specific codon optimality. This manuscript shows the interesting trends of tRNA modification changes and the corresponding enzyme expression. However, there are still many points that should be addressed before the publication.

Major comments:

1. The authors show attractive results in each figure, including tRNA epi-transcriptome changes. However, the relationships between each finding and viral expression are missing. Authors should perform ribosome profiling of virus-infected samples to investigate whether or not these RNA modification changes associate with smooth ribosome elongation on virus mRNAs and host mRNAs.
2. To investigate whether DNA damage response contributes to changes in tRNA modification under viral infections, authors should check whether artificially induced DNA damage by chemicals (without viral infection) shows reproducible epi-transcriptome results similar to viral infections, and confirm that DNA damage inhibitors counteract the tRNA modifications by LC-MS/MS and tRNA-seq.
3. To uncover the relationship between upregulated expression of KIAA1456/ALKBH8, mcm5U modification, codon optimality, and viral protein expression under viral infection, the authors should perform KIAA1456/ALKBH8 knockdown, check mcm5U modification, viral protein expression changes by Western blotting, and calculate ribosome occupancy across codons by ribosome profiling.
4. Similarly, to check the contribution of SLFN11/12 to reducing specific tRNA abundance, the authors should perform SLFN11/12 knockdown, check tRNA abundance, and perform ribosome profiling to calculate ribosome occupancy on target codons.
5. To confirm the relationship between oxidative stress, tRNA modification, and codon optimality, the authors should chemically induce oxidative stress, examine its effects on enzyme expression regulation and changes in tRNA modifications such as queuosine and f5C, and check the codon optimality by ribosome profiling.
6. The authors suggested that the protein level of the corresponding tRNA modification enzymes was changed upon viral

infection and the associated tRNA modification per se. This reviewer guessed that the enzyme expression comes first, but the mechanism was not studied. The authors should reveal the molecular mechanisms.

Minor comments:

1. Some Western blot results are ambiguous; the authors should repeat the blotting experiments. Figure 2B: pATM, ATM, and pATR bands are unclear, and GAPDH intensity appears to be saturated. Figure 6A: SLFN11/12 intensity appears to be saturated. The quantification of Western blots from triplicate (or more) would be helpful. Additionally, this reviewer wondered why there are two GAPDH bands in Figure 6A/B. Please explain the details.
2. The authors should check whether DDR inhibitors and/or oxidative stress inhibitors downregulate MAPK phosphorylation under viral infection.
3. *Homo sapiens* should be italicized in Figure 1 legends and Extended Data 1 legends.
4. The text in the heatmap of Extended Data 6 is too small and needs adjustment for readability.
5. The conclusion sentence in lines 172-174 was too strong from the data. The authors should tone down the nuance.
6. The sentence in lines 214-217, the authors explained that f5C was found in mt-tRNAs. This was indeed true for Q. This point should be highlighted.
7. The authors should discuss manQ and galQ as well.

Reviewer #4

(Remarks to the Author)

Reviewer #5

(Remarks to the Author)

Version 1:

Reviewer comments:

Reviewer #1

(Remarks to the Author)

The authors have addressed the majority of my concerns, as well as those raised by the other reviewers, and have significantly improved the manuscript. This reviewer greatly appreciates the approach of knocking down relevant modification enzymes to demonstrate a cause-effect-relationship between tRNA modification patterns and viral protein synthesis. The observation that knockdown of ELP3 and QTRT1, as well as overexpression of ADAT2 and ALKBH1, reduces viral protein production is highly interesting. To fully address my concern the authors should make an effort to better visualize the results presented in Figure 4. Quantification of knockdown/overexpression efficiency and the corresponding effects on viral protein levels would greatly help the reader. It should also be noted that some knockdown effects appear quite subtle, raising doubts about their impact on the associated tRNA modifications. I suggest that, for at least one enzyme, the authors provide targeted tRNA modification analysis to demonstrate a reduction in the relevant modifications following knockdown.

Reviewer #2

(Remarks to the Author)

We have completed our review of the revised manuscript. The authors have been responsive to the comments and have adequately addressed the concerns raised in our previous review. The additional experiments/analyses and clarifications have strengthened the manuscript. We believe the revised version is now suitable for publication.

Reviewer #3

(Remarks to the Author)

This reviewer thanks the authors for their extensive revisions and thoughtful responses to this reviewer's comments. The manuscript has been substantially improved. This reviewer appreciates the extensive efforts involved. The addition of enzyme perturbation experiments strengthens the work. However, this reviewer still has remaining concerns, primarily regarding the ribosome profiling analysis.

This reviewer appreciates the addition of enzyme knockdown/overexpression experiments and the codon frequency analysis shown in Extended Data 5. These are valuable data. However, this reviewer still respectfully and strongly requests doing ribosome profiling analysis.

The authors attribute their decision to omit ribosome profiling to technical limitations. However, this reviewer finds that this justification is premature. The authors cite a study (Finkel et al., 2021) as evidence that ribosome profiling at late infection stages is technically problematic due to loss of footprint periodicity on viral RNAs. However, another study (Kim et al., 2021), whose dataset the authors themselves use for the codon frequency analysis in Extended Data 5, appears to show clear 3-nucleotide periodicity at 36 hours post-infection. Thus, there is no technical hurdle for doing ribosome profiling in the author's condition, according to the literature.

This reviewer recommended doing the new ribosome profiling experiments that matched the authors' tRNA-Seq/mass spec data. Alternatively, the authors could reanalyze the Kim et al. dataset to calculate ribosome occupancy on each codon (viral and host mRNAs) (not the codon frequency analyzed in the Extended Data 5) and test the correspondence with tRNA modification changes (e.g., Q-modified, mcm5U-modified, and I-modified tRNAs).

For viral mRNAs, this reviewer understands the authors' concerns about non-ribosomal RNP complexes contributing to background signal, as noted in previous studies (Finkel et al., 2021; Jungfleisch et al., 2022). However, analytical strategies may address this issue. For instance, the authors could filter footprints by length, retaining only fragments in the 28-30 nucleotide range that match the canonical ribosome footprint size observed on host mRNAs. Alternatively, codon-specific metagene profiling, which averages ribosome occupancy around specific codons across many transcript instances, can reveal genuine translation signals even when background noise is present. Comparing such metagene profiles between mock-infected and infected samples (24-36 hpi) for codons affected by Q, mcm5U, and I modifications would directly test whether tRNA modification changes influence ribosome transit. This reviewer encourages the authors to explore whether such approaches could be applied.

The authors propose that tRNA-modifying enzymes are regulated post-transcriptionally during infection, based on decreased mRNA but increased protein levels (Figure 3G-J, Extended Data 6). This claim could be directly validated using the Kim et al. (2021) ribosome profiling dataset. Specifically, this reviewer asks the authors to analyze the translation efficiency (TE, calculated as ribosome-protected fragment reads / mRNA reads) of the key tRNA-modifying enzyme mRNAs—ELP3, QTRT1, ADAT2, ALKBH1, and NSUN2/3—across the infection time course to check the consistency with TE results in Extended Data 6.

Minor points

Line 136-139: citation of the paper is missing.

Reviewer #4

(Remarks to the Author)

Reviewer #5

(Remarks to the Author)

Version 2:

Reviewer comments:

Reviewer #1

(Remarks to the Author)

The authors have adequately addressed my issues and have improved the manuscript. I do not have additional comments.

Reviewer #3

(Remarks to the Author)

The authors answered the remaining concerns of this reviewer. The one note could be for the new sentence of "Moreover, ribosome occupancy analyses confirmed faster translation elongation at infection-induced tRNA-modification-dependent codons in TA transcripts relative to TR ones (Extended Data 5 D-F)" in lines 300 -302. If this reviewer's understanding is correct, the analysis in Extended Data 5D-F did not focus on the specific subgroup of mRNAs, but rather global changes in all the mRNAs in the dataset. Thus, the phrase regarding TA transcripts and TR transcripts in the corresponding sentence should be irrelevant. The authors should carefully revisit their own analysis and description of the data.

Also, the sentence summarized that all the corresponding codons provided faster elongation. However, this may not apply to some of the codons. The authors should rewrite this part more concretely, not exaggerating the results.

Reviewer #4

(Remarks to the Author)

POINT BY POINT RESPONSE

Reviewer #1 (Remarks to the Author):

In their manuscript “Coronaviruses reprogram the tRNA epitranscriptome to favor viral protein expression.” The authors demonstrate that coronavirus genomes are enriched with suboptimal A- and U-ending codons. They propose that this bias reflects an adaptation of viral codon usage to the host tRNA modification landscape under stress conditions induced by infection. The observations and underlying hypothesis are intriguing and may exemplify how the viral transcriptome has coevolved with the host tRNA modification machinery to optimize viral translation efficiency. However, the manuscript relies entirely on observational data and correlations, lacking direct cause-and-effect evidences and experimental validation. This limitation significantly weakens the strength and overall impact of the study.

Major:

POINT #1

To validate their hypothesis, the authors should consider manipulating the host cell tRNA modification machinery, using host cells that either lack or overexpress the tRNA modification enzymes implicated in viral adaptation. One might expect reduced viral protein expression in knockout cells deficient in specific enzymes, such as Qtrt1/Qtrt2, as well as in cells overexpressing ADAT2. The same principle applies to NSun2 and ALKBH8.

ANSWER #1:

To address this key issue, we focused on HCoV-OC43 infection on A549 cells because **(i)** the observed changes in tRNA modifications were conserved across both viruses analyzed, **(ii)** transfection efficiency in the Calu3 and MRC5 cells lines (used for SARS CoV-2 and HCoV-OC43 infection, respectively) is poor and **(iii)** A549 cells support HCoV-OC43 replication (**Extended Data 3D-F**) and can be efficiently transfected.

Consistent with a functional causal link between the observed tRNA epitranscriptome alterations and viral protein synthesis, we found that **(i)** siRNA-mediated knockdown of the tRNA modifying enzymes ELP3 (mcm⁵U modification pathway) and QTRT1 (queuosine modification pathway), as well as **(ii)** overexpression of ADAT2 (A-to-I modification enzyme) and ALKBH1 (f⁵C pathway), each significantly reduced viral protein levels (N-protein). In contrast, NSUN2 silencing had no detectable effect. The lack of an effect upon NSUN2 depletion, combined with the observed downregulation of ALKBH1 during infection and the strong inhibitory effect on viral N-protein expression upon its overexpression, suggests that the virus primarily modulates the NSUN3–ALKBH1 mitochondrial pathway, rather than the cytosolic NSUN2-dependent m⁵C modification. This interpretation is further supported by the lack of detectable changes in global m⁵C tRNA modification levels during infection (**Figure 3A**).

We silenced ELP3 instead of KIAA1456 or ALKBH8 because it acts upstream in the mcm⁵ modification pathway, directly influencing the activity of these enzymes, and because ELP3 expression itself was strongly upregulated during HCoV-OC43 infection (**Figure 4A**).

Together, these findings demonstrate that the observed alterations of the tRNA-modifying enzymes under infection conditions functionally contribute to coronavirus viral protein production. These results are now included in the new Figure 4 of the revised manuscript. The following text has been added at lines 273 - 291:

“To establish a causal link between the alteration of these enzymes and viral protein expression, we manipulated the expression of selected tRNA modifying enzymes. These experiments were performed using HCoV-OC43 infection in A549 cells because the observed changes in tRNA modifications were conserved across both viruses analyzed, transfection efficiency in Calu3 and MRC5 cell lines (used for SARS-CoV-2 and HCoV-

OC43 infection, respectively) is poor, and A549 cells support efficient HCoV-OC43 replication and transfection (Extended Data Figure 3D-F). Consistent with a functional link between tRNA epitranscriptome remodeling and viral protein production, siRNA-mediated knockdown of ELP3 (mcm⁵U modification pathway) and QTRT1 (queuosine modification pathway) significantly reduced viral protein levels (Figure 4A-B). ELP3 was selected for silencing because it acts upstream of KIAA1456 and ALKBH8 in the mcm⁵U modification pathway and is strongly upregulated during HCoV-OC43 infection (Figure 4A). Similarly, overexpression of ADAT2 (A-to-I modification enzyme) or ALKBH1 (f⁵C pathway) also led to a pronounced decrease in viral protein expression (Figure 4D). In contrast, NSUN2 silencing had no detectable effect (Figure 4C). This lack of effect, together with the observed downregulation of ALKBH1 during infection (Figure 3H) and the strong inhibition of viral protein production upon its overexpression, suggests that viral infection primarily modulates the mitochondrial NSUN3-ALKBH1 pathway rather than the cytosolic NSUN2-dependent m⁵C modification. This interpretation is further supported by the absence of significant changes in global tRNA m⁵C levels analyzed by LC-MS/MS during infection (Figure 3A). Together, these findings demonstrate that alterations in mcm⁵U, Q and I tRNA-modifying enzymes during infection not only align with viral codon usage but also functionally contribute to coronavirus protein production.”

POINT #2

Mass spectrometry reveals variations in overall modifications upon infection but fails to demonstrate site-specific changes. While some observed trends align with the author’s hypothesis, it remains unclear whether the changes are occurring on the expected tRNAs at position N34 and whether they result from an increase or decrease in modifications at these sites. The observed changes may be due to variations in tRNA expression, as suggested in Figure 5, which would inevitably affect the modification-to-tRNA ratio. To validate conclusions regarding changes in tRNA modifications upon viral infection, a modification/tRNA-targeted approach is necessary.

ANSWER #2:

We thank the reviewer for the comment and apologize for any lack of clarity in our initial text. We agree that certain tRNA modifications can occur at multiple positions. However, f mcm⁵U/mcm⁵s²U, queuosine (Q), inosine (I), and f⁵C modification analyzed in our study are well established to occur exclusively at position N34 of their cognate tRNAs, as described in Suzuki T., 2021 (See Reference 1 listed at the end of this response). The only exception was m⁵C that can occur at multiple positions. Moreover, to exclude the possibility that the observed differences resulted from altered tRNA expression rather than modification levels, we performed mim-tRNAseq analysis. Differential expression analysis (now Figure 6A-F) revealed no significant changes in the abundance of tRNAs carrying these modifications. We have now adjusted the sentence at lines 128 – 130:

“Indeed, the decoding of 48% of codons enriched in coronavirus genomes is affected by four tRNA modifications: I, Q, mcm⁵/mcm⁵s²U, which occur exclusively at position N34, and m⁵C, which can also occur at other tRNA positions besides N34”.

Moreover, we have included a final sentence at lines 364–366 to further clarify this point.

“Importantly, the tRNAs exhibiting altered expression levels are distinct from those displaying modification changes, indicating that abundance and modification are regulated through separate mechanisms”.

1. Suzuki, T. The expanding world of tRNA modifications and their disease relevance. *Nat Rev Mol Cell Biol* **22**, 375–392 (2021). <https://doi.org/10.1038/s41580-021-00342-0>.

POINT #3

Moreover, since mitochondrial tRNA modifications should not directly affect viral protein translation, their contribution, along with mit-tRNA expression, to the overall observed tRNA modification changes remains unclear.

ANSWER #3:

We agree with the reviewer that mitochondrial tRNA modifications are unlikely to directly influence viral protein translation. In this study, we did not specifically dissect the contribution of mitochondrial tRNAs or mitochondrial processes. However, we note that one of the modifications altered during infection, 5-formylcytidine (f⁵C), occurs in both cytoplasmic (tRNA^{LeuCAA}) and mitochondrial (tRNA^{iMet}) tRNAs. Given this dual localization, the changes observed in f⁵C levels may reflect infection-induced mitochondrial stress. Supporting this idea, as part of a new ongoing project, we have found comparable f⁵C alterations during infections with viruses exhibiting optimal codon usage, suggesting that mitochondrial perturbations may contribute to these modification changes independently of viral codon usage bias. We will now clarify and discuss this point in the revised manuscript by adding the sentence at lines 222-224:

“Given that f⁵C is present in both cytoplasmic and mitochondrial tRNAs, and LC-MS/MS analysis cannot discriminate between cytosolic and mitochondrial tRNA pools, the observed f⁵C changes may reflect mitochondrial stress responses rather than direct effects on viral protein translation”.

POINT #4

Even if some modification derivatives are evaluated and discussed, others like Q/manQ/galQ (PMID: 37992713) are completely ignored, especially in their impact on the viral codon translation.

ANSWER #4:

We thank the reviewer for pointing this out. We have now included the following text in the revised manuscript addressing the potential impact of manQ and galQ on viral codon translation at lines 136 – 139:

“The Q modification and its derivatives, mannosyl-Q (manQ) and galactosyl-Q (galQ), favor the decoding of four U-ending codons (Asparagine (Asn) AAU, Aspartic acid (Asp) GAU, Histidine (His) CAU, and Tyrosine (Tyr) UAU), with galQ found in human tRNA^{Tyr}, and manQ found in human tRNA^{Asp} (Figure 1D)”.

Minor:

POINT #5

Page 4 row 136 the statement “ When Q-modified, G34, which normally pairs with C or with near-cognate Q-decoded codons (such as Glu-GAA and Glu-GAG), can also base pair with U, disfavoring decoding of coronavirus codons” is in contradiction with Figure 1F. Please clarify.

ANSWER #5:

We thank the reviewer for noting this mistake. The wording in the manuscript was incorrect and should read “**favoring**” instead of “**disfavoring**.” We have corrected this in the revised version.

Reviewer #2 (Remarks to the Author):

In this study, authors found that SARS-CoV-2 and HCoV-OC43 infection induces extensive reprogramming of inosine, queuosine, mcm5U/mcm5s2U and m5C/f5C tRNA modifications. It is an interesting study that contributes to our understanding of host-virus interactions, particularly how viruses can exploit epitranscriptome mechanisms to improve viral protein expression. However, there are several specific

concerns and results interpretations that need to be addressed to support stated conclusions and strengthen the significance of these findings.

Major Comments

POINT #1

The authors used p-p38 to analyze oxidative stress. The activation of p38 MAPK is involved in various biological processes. Therefore, p-p38 is not an ideal biomarker of oxidative stress. The authors should provide ROS or MDA analysis data to confirm the oxidative stress.

ANSWER #1:

We thank the reviewer for the comment. To directly address oxidative stress, we performed a DCFH-DA assay to measure ROS production in HCoV-OC43 infected cells at 48 hpi compared with mock-infected controls. The results show a significant increase in ROS levels upon infection. As a positive control, H₂O₂ treatment produced the expected ROS induction. These new data have been included in the new **Extended Data 2A-B** of the revised manuscript and the following part has been added to the text at lines 177 -178.

*“Furthermore, ROS production was confirmed by directly measuring ROS levels in infected cell (**Extended Data 2A-B**)”.*

POINT #2

The authors used MOI 3 for SARS-CoV-2 infection, MOI 0.1 for OC43 infection. Why use such a high MOI for SARS-CoV-2? Are similar tRNA changes observed at MOI 0.1? If yes, please provide the results from low MOI to higher MOI conditions. Were these changes only observed under high MOI conditions? If yes, why?

ANSWER #2:

The MOI values were chosen based on the replication kinetics of each virus in the specific cell lines used. Our goal was to achieve robust infection while minimizing cytotoxic effects, ensuring that observed tRNA modification changes reflected biological regulation rather than cell death.

For SARS-CoV-2, an MOI of 3 in Calu-3 cells is commonly used in the literature and provides consistent replication within 30–48 hours post-infection. Lower MOI values in this cell line result in markedly delayed replication and require prolonged incubation to reach comparable infection levels, potentially introducing confounding variables. In contrast, for HCoV-OC43, an MOI of 0.1 was sufficient for efficient infection, as higher MOIs did not further increase viral replication, likely due to differences in entry efficiency or host cell permissiveness.

POINT #3

In figure 3, authors showed the up/downregulation of inosine, queuosine, mcm5U, and f5C in cells. However, only the increase of mcm5U was further confirmed *in vivo*. The other tRNA changes also need to be confirmed *in vivo*.

ANSWER #3:

In our *in vivo* analyses, we examined the complete panel of tRNA modifications. However, only mcm⁵U showed a statistically significant increase. We attribute this to the low, partial and heterogeneous infection levels obtained *in vivo*, where it is not possible to selectively analyze highly infected regions. Consistent with this, variability among samples allowed us to distinguish two subgroups based on infection levels. Despite this limitation, we confirmed the functional relevance of tRNA modifications through the newly added siRNA knockdown and overexpression experiments in cell culture, which demonstrated clear effects on viral protein expression (new **Figure 4**). We agree that more refined *in vivo* infection models could provide further

information into the regulation of other tRNA modifications, and we highlight this as an important direction for future studies. We have now included these sentences to clarify our *in vivo* findings at lines 404 – 410:

“Upregulation of mcm⁵U was observed in both cell culture and in hamsters infected with the SARS-CoV-2 ancestral strain D641G, which has lower infectivity compared to the SARS-CoV-2 strain used in our in vitro experiments. This upregulation was higher in samples with the highest viral load, suggesting that infection efficiency is a critical factor influencing the levels of tRNA modifications (Figure 3E). Taken together, these findings suggest that infection efficiency influences tRNA modification levels, which might explain why the other tRNA modifications were not altered in vivo. Further in vivo experiments with a more suitable animal model will be required to clarify this matter”.

POINT #4

In figure3, the OC43 infection induced more extensive tRNA modification changes than SARS-CoV-2. However, in figure5, the host tRNA abundance only was observed in SARS-COV-2 infection but not in OC43 infection. Is it reasonable for such a phenomenon to occur? Please clarify this point. And the title “Coronaviruses alter the host tRNA abundance” of this section should be changed.

ANSWER #4:

We attribute the differences in tRNA abundance changes between SARS-CoV-2 and OC43 infections to distinct host responses in the respective cell lines used. The cell line used for SARS-CoV-2 but not the one used OC43 infection is capable of mounting a robust type I interferon (IFN) response upon infection. IFN is well established to induce SLFN11 and SLFN12 expression, two proteins known to cleave type II tRNAs (including tRNA^{Ser} and tRNA^{Leu}), the ones we observed to be downregulated, as part of the antiviral defense mechanism (See References 1–5 listed at the end of this response).

We agree that the current section title is too general. To better reflect our key findings, we have revised it to “**Coronaviruses reprogram modifications in pre-existing tRNAs without altering their abundance**” to highlight this aspect in the results.

The differences in tRNA modification and tRNA abundance changes between SARS-CoV-2 and OC43 infection are addressed at lines 342-350 and 427-430.

“Regarding the downregulated tRNAs, all belong to the type II tRNAs, which are substrates for Schlafen 11 (SLFN11) and Schlafen 12 (SLFN12) endonucleases⁴⁴. These enzymes are induced by the interferon (IFN) response as part of the innate antiviral defense or in response to DNA damage^{45–48}. Consistent with the downregulation of type II tRNAs in SARS-CoV-2 infection, but not in HCoV-OC43 infection, SARS-CoV-2 infection triggered an IFN response, as indicated by the expression of IFN-stimulated genes and the strong upregulation of SLFN11 and SLFN12 (Extended Data 9A-C). In contrast, HCoV-OC43 did not induce a proper IFN response and only led to a slight activation of IFNB1 at 72 hours post-infection, likely due to the DNA damage response (Extended Data 9B-D). These differences are linked to the cell lines used for SARS-CoV-2 and HCoV-OC43 experiments”

“The differences in tRNA modification profiles observed between SARS-CoV-2 and HCoV-OC43 infections may stem from the distinct cell lines used, the virus themselves, or both. SARS-CoV-2 infections were conducted in the lung adenocarcinoma Calu3 cell line, whereas HCoV-OC43 infections were conducted in the lung fibroblast MRC5 cell line, both well-established and recommended models for their study. Cancer cell lines have been reported to exhibit altered levels of tRNA modifications, which may influence virus-induced tRNA modification profiles”

References:

1. Kobayashi-Ishihara, M., Frazão Smutná, K., Alonso, F.E. *et al.* Schlafen 12 restricts HIV-1 latency reversal by a codon-usage dependent post-transcriptional block in CD4+ T cells. *Commun Biol* **6**, 487 (2023). <https://doi.org/10.1038/s42003-023-04841-y>.
2. Li, M., Kao, E., Malone, D. *et al.* DNA damage-induced cell death relies on SLFN11-dependent cleavage of distinct type II tRNAs. *Nat Struct Mol Biol* **25**, 1047–1058 (2018). <https://doi.org/10.1038/s41594-018-0142-5>
3. Metzner, F.J., Wenzl, S.J., Kugler, M. *et al.* Mechanistic understanding of human SLFN11. *Nat Commun* **13**, 5464 (2022). <https://doi.org/10.1038/s41467-022-33123-0>
4. Kim, E. T. & Weitzman, M. D. Schlafens Can Put Viruses to Sleep. *Viruses* **14**, 442 (2022). <https://doi.org/10.3390/v14020442>
5. Valdez F. *et al.* . Schlafen 11 Restricts Flavivirus Replication. *J Virol* 2019 93:10.1128/jvi.00104-19. <https://doi.org/10.1128/jvi.00104-19>

POINT #5

In Figure 6, it is unclear whether the same MOIs were used as in previous experiments—MOI 3 for SARS-CoV-2 and MOI 0.1 for OC43. If this is the case, the observed differences in SLFN11/12 expression and interferon-related genes expression between the two viruses may be influenced. The authors should clarify the MOIs used in this figure and discuss how differences in viral dose may impact the interpretation of host response data.

ANSWER #5:

Thank you for pointing this out. The MOIs used in Figure 6 (now **Extended Data Figure 9**) were identical to those applied in the previous experiments, as they were carefully optimized for each virus to ensure efficient infection while preserving cell viability in the respective cell lines. We have now clarified this point in the figure legend of **Extended Data Figure 9**. We attribute the observed differences in SLFN11/12 and interferon-related gene expression to the intrinsic responsiveness of the cell lines to IFN induction rather than to differences in viral dose.

Minor Comments

POINT #6

The title claims that coronaviruses reprogram the tRNA epitranscriptome to favor viral protein expression. Therefore, the increase of viral protein levels is a critical phenomenon. The authors should show viral protein expression levels in main figures not just in supplementary figures. Extended figure 2 showed only SARS-COV-2 protein levels. The authors should add OC43 viral protein expression data.

ANSWER #6:

In the revised manuscript, we have now included a new figure showing a time-course analysis of OC43 viral protein expression in both MRC5 and A549 cells. Due to space constraints, these data were placed in the supplementary section (**Extended Data 3F**). However, the effects of siRNA knockdown and overexpression of tRNA-modifying enzymes on viral protein expression are presented in the main figures, as these directly support our central conclusion that tRNA modifications regulate viral protein production (**Figure 4**).

POINT #7

The authors propose that SLFN11/12 upregulation leads to a decrease in type 2 tRNA abundance under SARS-COV-2 infection. However, the observed timing is inconsistent: the reduction in type 2 tRNAs occurs at 32 hours post-infection, whereas SLFN11/12 expression increases later, at 48 hours. This temporal discrepancy raises questions about whether SLFN11/12 induction is the cause of the tRNA reduction or a downstream

consequence. Clarification of this timeline, or additional data supporting a delayed effect of SLFN11/12, would strengthen the authors' conclusion.

ANSWER #7:

In the revised manuscript, we have included a replicate experiment in which the Western blots signals were acquired under non-saturating conditions, allowing proper quantification (new **Extended Data 9A and B**). These analyses show that SLFN11/12 expression begins to increase as early as 24 hours post-infection and then gradually decline at later time points. This revised analysis aligns with the observed reduction in type II tRNAs at 32 hours post-infection, supporting the conclusion that SLFN11/12 upregulation contributes to the observed decrease in tRNA abundance.

POINT #8

Add a graphical abstract or summary figure may help readers grasp the key findings.

ANSWER #8:

Thank you for the suggestion, we have now added a graphical summary in the revised manuscript, **Extended Data 11B**.

Reviewer #3 (Remarks to the Author):

Coronavirus genomes are enriched in suboptimal codons, which potentially reduce the rate of protein synthesis. However, coronavirus proteins can be efficiently translated despite the codon usage conflict. To tackle this inconsistency, the authors uncovered that coronavirus infection changes tRNA-modifying enzyme expression and tRNA epi-transcriptomics to modulate specific codon optimality. This manuscript shows the interesting trends of tRNA modification changes and the corresponding enzyme expression. However, there are still many points that should be addressed before the publication.

Major comments:

POINT #1

The authors show attractive results in each figure, including tRNA epi-transcriptome changes. However, the relationships between each finding and viral expression are missing. Authors should perform ribosome profiling of virus-infected samples to investigate whether or not these RNA modification changes associate with smooth ribosome elongation on virus mRNAs and host mRNAs.

ANSWER #1:

We fully agree that ribosome profiling could provide important insights into how tRNA modification changes influence translational elongation. However, under our infection conditions-when viral replication and translation are highly active, accurate ribosome profiling faces major technical limitations. As shown in previous studies (**See References 1–3 listed at the end of this response**) the characteristic ribosome footprint periodicity is lost in viral RNAs due to the accumulation of non-ribosomal, RNase-resistant RNA fragments. These fragments likely arise from the association of multiple host and viral proteins with viral RNAs, masking genuine ribosome footprints and obscuring codon-level analyses. Consequently, ribosome profiling of coronaviruses (including SARS-CoV-2) is typically limited to early infection stages, when such artifacts are minimal (**See References 1 and 2 listed at the end of this response**). However, early-stage infections do not capture the extensive host perturbations occurring during high replication conditions, the context in which we observe infection-induced tRNA modification changes. To address the reviewer's request and functionally link tRNA modification changes with translational regulation, we performed two complementary analyses:

1. Functional validation through enzyme perturbation:

To address this key issue, we focused on HCoV-OC43 infection on A549 cells because **(i)** the observed changes in tRNA modifications were conserved across both viruses analyzed, **(ii)** transfection efficiency in the Calu3 and MRC5 cells lines (used for SARS CoV-2 and HCoV-OC43 infection, respectively) is poor and **(iii)** A549 cells support HCoV-OC43 replication (**Extended Data 3D-F**) and can be efficiently transfected.

Consistent with a functional causal link between the observed tRNA epitranscriptome alterations and viral protein synthesis, we found that **(i)** siRNA-mediated knockdown of the tRNA modifying enzymes ELP3 (mcm⁵U modification pathway) and QTRT1 (queuosine modification pathway), as well as **(ii)** overexpression of ADAT2 (A-to-I modification enzyme) and ALKBH1 (f⁵C pathway), each significantly reduced viral protein levels (N-protein). In contrast, NSUN2 silencing had no detectable effect. The lack of an effect upon NSUN2 depletion, combined with the observed downregulation of ALKBH1 during infection and the strong inhibitory effect on viral N-protein expression upon its overexpression, suggests that the virus primarily modulates the NSUN3–ALKBH1 mitochondrial pathway, rather than the cytosolic NSUN2-dependent m⁵C modification. This interpretation is further supported by the lack of detectable changes in global m⁵C tRNA modification levels during infection (**Figure 3A**). We silenced ELP3 instead of KIAA1456 or ALKBH8 because it acts upstream in the mcm⁵ modification pathway, directly influencing the activity of these enzymes, and because ELP3 expression itself was strongly upregulated during HCoV-OC43 infection (**Figure 4A**). Together, these findings demonstrate that the observed alterations of the tRNA-modifying enzymes under infection conditions functionally contribute to coronavirus viral protein production. These results are now included in the new **Figure 4** of the revised manuscript. The following text has been added to the manuscript at lines 273-291:

*“To establish a causal link between the alteration of these enzymes and viral protein expression, we manipulated the expression of selected tRNA modifying enzymes. These experiments were performed using HCoV-OC43 infection in A549 cells because the observed changes in tRNA modifications were conserved across both viruses analyzed, transfection efficiency in Calu3 and MRC5 cell lines (used for SARS-CoV-2 and HCoV-OC43 infection, respectively) is poor, and A549 cells support efficient HCoV-OC43 replication and transfection (**Extended Data Figure 3D-F**). Consistent with a functional link between tRNA epitranscriptome remodeling and viral protein production, siRNA-mediated knockdown of ELP3 (mcm⁵U modification pathway) and QTRT1 (queuosine modification pathway) significantly reduced viral protein levels (**Figure 4A–B**). ELP3 was selected for silencing because it acts upstream of KIAA1456 and ALKBH8 in the mcm⁵U modification pathway and is strongly upregulated during HCoV-OC43 infection (**Figure 4A**). Similarly, overexpression of ADAT2 (A-to-I modification enzyme) or ALKBH1 (f⁵C pathway) also led to a pronounced decrease in viral protein expression (**Figure 4D**). In contrast, NSUN2 silencing had no detectable effect (**Figure 4C**). This lack of effect, together with the observed downregulation of ALKBH1 during infection (**Figure 3H**) and the strong inhibition of viral protein production upon its overexpression, suggests that viral infection primarily modulates the mitochondrial NSUN3–ALKBH1 pathway rather than the cytosolic NSUN2-dependent m⁵C modification. This interpretation is further supported by the absence of significant changes in global tRNA m⁵C levels analyzed by LC-MS/MS during infection (**Figure 3A**). Together, these findings demonstrate that alterations in mcm⁵U, Q and I tRNA-modifying enzymes during infection not only align with viral codon usage but also functionally contribute to coronavirus protein production.”*

Together, these results indicate that coronavirus infection remodels the host tRNA epitranscriptome modulating codon optimality and enhancing viral protein translation. This functional relationship provides a mechanistic link between tRNA modification dynamics and viral gene expression, even in the absence of technically feasible ribosome profiling at peak replication stages.

2. Codon optimality analysis:

We reasoned that the infection-induced tRNA modifications would cause a change in codon optimality that should impact both viral and host mRNAs. Accordingly, we compared the codon composition of translation-activated (TA) and translation-repressed (TR) host mRNAs under SARS-CoV-2 infection published by *Kim D. et al* (**See Reference 4 listed at the end of this response**). We found that TA mRNAs were enriched in codons preferentially decoded by tRNAs bearing an I-unmodified (A-ending tRNAs), mcm⁵U and Q modifications, while TR mRNAs were enriched in codons preferentially decoded by I-modified tRNAs. These results are consistent with SARS-CoV-2 codon preferences and indicate that infection drives a shift in codon optimality through remodeling of the tRNA epitranscriptome that favor viral protein synthesis. The corresponding analyses are now presented in **Extended Data 5** and the following text has been added to the manuscript at lines 292 – 299:

*“To further confirm the link between coronaviruses infection, tRNA modifications and viral protein expression, we analyzed codon frequency in host transcripts translation-activated (TA) versus translation-repressed (TR) in ribosome profiling datasets from SARS-CoV-2–infected Calu3 cells. We reasoned that coronavirus-induced alterations in tRNA modifications would modify codon optimality, thereby influencing not only viral but also host mRNA translation. Consistently, we found that, similar to coronavirus mRNAs, host TA mRNAs were enriched in codons preferentially decoded by tRNAs bearing an I-unmodified, mcm⁵U and Q modifications, whereas TR mRNAs were enriched in codons preferentially decoded by I-modified tRNAs (**Extended Data 5**)”.*

References:

1. Finkel, Y., Mizrahi, O., Nachshon, A. *et al.* The coding capacity of SARS-CoV-2. *Nature* **589**, 125–130 (2021). <https://doi.org/10.1038/s41586-020-2739-1>;
2. Finkel, Y., Gluck, A., Nachshon, A. *et al.* SARS-CoV-2 uses a multipronged strategy to impede host protein synthesis. *Nature* **594**, 240–245 (2021). <https://doi.org/10.1038/s41586-021-03610-3>;
3. Jungfleisch, J., Böttcher, R., Talló-Parra, M. *et al.* CHIKV infection reprograms codon optimality to favor viral RNA translation by altering the tRNA epitranscriptome. *Nat Commun* **13**, 4725 (2022). <https://doi.org/10.1038/s41467-022-31835-x>.
4. Kim D. *et al.* “A high-resolution temporal atlas of the SARS-CoV-2 translome and transcriptome”. *Nat Commun* **12**, 5120 (2021) <https://doi.org/10.1038/s41467-021-25361-5>”

POINT #2

To investigate whether DNA damage response contributes to changes in tRNA modification under viral infections, authors should check whether artificially induced DNA damage by chemicals (without viral infection) shows reproducible epi-transcriptome results similar to viral infections, and confirm that DNA damage inhibitors counteract the tRNA modifications by LC-MS/MS and tRNA-seq.

ANSWER #2:

We apologize for the lack of accuracy in the text. Indeed, several previous studies have extensively shown that the tRNA epitranscriptome is tightly regulated under stress conditions, including the DNA damage response. The DNA damage response genes are enriched, as coronaviruses, in codons whose decoding is favored by mcm⁵ modification in the counterpart tRNAs. We have now revised and updated the list of references (**See References 1–7 listed at the end of this response**).

References:

1. Mitchener M.M., Begley T.J., Dedon P.C. Molecular Coping Mechanisms: Reprogramming tRNAs To Regulate Codon-Biased Translation of Stress Response Proteins. *Acc Chem Res.* 2024 Aug 20;57(16):2448. <https://doi.org/10.1021/acs.accounts.3c00572>;
2. Chan C. *et al.* Reprogramming of tRNA modifications controls the oxidative stress response by codon-biased translation of proteins. *Nat Commun.* **3**, 937 (2012). <https://doi.org/10.1038/ncomms1938>;
3. Begley U. *et al.* Trm9-catalyzed tRNA modifications link translation to the DNA damage response. *Mol. Cell.* 2007 Dec 14;28(5):860-70. <https://doi.org/10.1016/j.molcel.2007.09.021>;
4. Gu C., Begley T.J., Dedon P.C. tRNA modifications regulate translation during cellular stress. *FEBS Lett.* 2014 Nov 28;588(23):4287-96. <https://doi.org/10.1016/j.febslet.2014.09.038>;
5. Huber S.M. *et al.* Arsenite toxicity is regulated by queuine availability and oxidation-induced reprogramming of the human tRNA epitranscriptome. *Proc Natl Acad Sci U S A.* 2022 Sep 20;119(38):e2123529119. <https://doi.org/10.1073/pnas.2123529119>;
6. Rashad, S. *et al.* Translational response to mitochondrial stresses is orchestrated by tRNA modifications. *bioRxiv* 2024.02.14.580389 (2024) <https://doi.org/10.1101/2024.02.14.580389>.
7. Patil A *et al.* Increased tRNA modification and gene-specific codon usage regulate cell cycle progression during the DNA damage response. *Cell Cycle.* 2012 Oct 1;11(19):3656-65. <https://doi.org/10.4161/cc.21919>.

POINT #3

To uncover the relationship between upregulated expression of KIAA1456/ALKBH8, mcm5U modification, codon optimality, and viral protein expression under viral infection, the authors should perform KIAA1456/ALKBH8 knockdown, check mcm5U modification, viral protein expression changes by Western blotting, and calculate ribosome occupancy across codons by ribosome profiling.

ANSWER #3:

Because of the ribosome profiling technical difficulties commented in *Answer #1*, we directly addressed the reviewer request by experimentally manipulating the relevant tRNA modification enzymes and testing the effect on viral protein levels, and testing the effect of viral infection on codon optimality. The results, already described in *Answer #1*, provide a functional link between the effect of SARS-CoV-2-induced tRNA epitranscriptome alterations and viral protein production.

POINT #4

Similarly, to check the contribution of SLFN11/12 to reducing specific tRNA abundance, the authors should perform SLFN11/12 knockdown, check tRNA abundance, and perform ribosome profiling to calculate ribosome occupancy on target codons.

ANSWER #4:

Both SLFN11 and SLFN12. Are well established interferon-stimulated genes that cleave type II tRNAs (**See References 1–5 listed at the end of this response**). Consistently, in our data we show that SARS-CoV-2 infection, performed in an IFN-competent cell line, shows a selective reduction of type-II tRNAs that matches these known SLFN-dependent cleavage patterns. By contrast, the HCoV-OC43 experiments were performed in an IFN-defective cell line, consistent with minimal SLFN induction and the absence of comparable tRNA losses. Together, these literature-supported mechanisms and the distinct innate capacities of the two cell systems provide the most plausible explanation for our observations. The manuscript now explicitly discusses this interpretation at 342-350 includes references to the foundational SLFN11/12 literature.

“Regarding the downregulated tRNAs, all belong to the type II tRNAs, which are substrates for Schlafen 11 (SLFN11) and Schlafen 12 (SLFN12) endonucleases⁴⁴. These enzymes are induced by the interferon (IFN) response as part of the innate antiviral defense or in response to DNA damage^{45–48}. Consistent with the

downregulation of type II tRNAs in SARS-CoV-2 infection, but not in HCoV-OC43 infection, SARS-CoV-2 infection triggered an IFN response, as indicated by the expression of IFN-stimulated genes and the strong upregulation of SLFN11 and SLFN12 (**Extended Data 9A-C**). In contrast, HCoV-OC43 did not induce a proper IFN response and only led to a slight activation of IFNB1 at 72 hours post-infection, likely due to the DNA damage response (**Extended Data 9B-D**). These differences are linked to the cell lines used for SARS-CoV-2 and HCoV-OC43 experiments.”

References:

1. Kobayashi-Ishihara, M., Frazão Smutná, K., Alonso, F.E. *et al.* Schlafen 12 restricts HIV-1 latency reversal by a codon-usage dependent post-transcriptional block in CD4+ T cells. *Commun Biol* **6**, 487 (2023). <https://doi.org/10.1038/s42003-023-04841-y>.
2. Li, M., Kao, E., Malone, D. *et al.* DNA damage-induced cell death relies on SLFN11-dependent cleavage of distinct type II tRNAs. *Nat Struct Mol Biol* **25**, 1047–1058 (2018). <https://doi.org/10.1038/s41594-018-0142-5>
3. Metzner, F.J., Wenzl, S.J., Kugler, M. *et al.* Mechanistic understanding of human SLFN11. *Nat Commun* **13**, 5464 (2022). <https://doi.org/10.1038/s41467-022-33123-0>
4. Kim, E. T. & Weitzman, M. D. Schlafens Can Put Viruses to Sleep. *Viruses* **14**, 442 (2022). <https://doi.org/10.3390/v14020442>
5. Valdez F. *et al.* Schlafen 11 Restricts Flavivirus Replication. *J Virol* 2019 93:10.1128/jvi.00104-19. <https://doi.org/10.1128/jvi.00104-19>

POINT #5

To confirm the relationship between oxidative stress, tRNA modification, and codon optimality, the authors should chemically induce oxidative stress, examine its effects on enzyme expression regulation and changes in tRNA modifications such as queuosine and f5C, and check the codon optimality by ribosome profiling.

ANSWER #5:

We thank the reviewer for this comment. The link between oxidative stress, tRNA modification, and codon-biased translation has been previously demonstrated as commented in *Answer #2 (See References 1–6 listed at the end of this response)*. We have now included the complete set of references

References:

1. Mitchener M.M., Begley T.J., Dedon P.C. Molecular Coping Mechanisms: Reprogramming tRNAs To Regulate Codon-Biased Translation of Stress Response Proteins. *Acc Chem Res.* 2024 Aug 20;57(16):2448. <https://doi.org/10.1021/acs.accounts.3c00572>;
2. Chan C. *et al.* Reprogramming of tRNA modifications controls the oxidative stress response by codon-biased translation of proteins. *Nat Commun.* **3**, 937 (2012). <https://doi.org/10.1038/ncomms1938>;
3. Begley U. *et al.* Trm9-catalyzed tRNA modifications link translation to the DNA damage response. *Mol. Cell.* 2007 Dec 14;28(5):860-70. <https://doi.org/10.1016/j.molcel.2007.09.021>;
4. Gu C., Begley T.J., Dedon P.C. tRNA modifications regulate translation during cellular stress. *FEBS Lett.* 2014 Nov 28;588(23):4287-96. <https://doi.org/10.1016/j.febslet.2014.09.038>;
5. Huber S.M *et al.* Arsenite toxicity is regulated by queuine availability and oxidation-induced reprogramming of the human tRNA epitranscriptome. *Proc Natl Acad Sci U S A.* 2022 Sep 20;119(38):e2123529119. <https://doi.org/10.1073/pnas.2123529119>;
6. Rashad, S. *et al.* Translational response to mitochondrial stresses is orchestrated by tRNA modifications. *bioRxiv* 2024.02.14.580389 (2024) <https://doi.org/10.1101/2024.02.14.580389>.

POINT #6

The authors suggested that the protein level of the corresponding tRNA modification enzymes was changed upon viral infection and the associated tRNA modification per se. This reviewer guessed that the enzyme expression comes first, but the mechanism was not studied. The authors should reveal the molecular mechanisms.

ANSWER #6:

We have now expanded our analyses to address temporal order and mode of regulation. Across MRC5 and A549 cells, RT-qPCR shows that tRNA-modifying enzyme mRNAs decrease during infection, whereas immunoblotting demonstrates a time-dependent increase in the corresponding proteins (24–72 hpi), indicating post-transcriptional upregulation rather than transcriptional induction (**Figure 3; Extended Data 6**). The following text has been added to the manuscript at lines **306-311**:

“Finally, to determine whether the expression of tRNA-modifying enzymes occurs at the transcriptional or translational level during infection, we analyzed their mRNA and protein abundance. qPCR analyses revealed that the mRNA levels were decreased (Extended Data 6A-F), whereas western blot analyses showed a concomitant increase in the protein levels (Figure 3G-J; Extended Data 6G). These results indicate that expression of tRNA-modifying enzymes is primarily regulated at the translational level rather than transcriptionally during infection”.

The rise in enzyme protein, precedes or coincides with the observed shifts in tRNA modification levels (**Figure 3**), supporting that enzyme accumulation drives the modification changes. Pharmacological inhibition of the DNA-damage response (DDR) reduces viral protein production (**Extended Data 2C**), placing enzyme upregulation downstream of virus-activated stress/DDR signaling (lines 170 – 177). Functionally, siRNA knockdown of key enzymes reduces viral protein accumulation (**Figure 4**), establishing their contribution to efficient infection (lines 273-291). Collectively, these data support a model in which virus-induced stress/DDR pathways promote translation of tRNA-modifying enzymes, reprogramming the tRNA-modification landscape, consequently enhancing viral protein synthesis. We have now included a Graphical Summary to illustrate our model (**Extended Data 11B**).

Minor comments:

POINT #7

Some Western blot results are ambiguous; the authors should repeat the blotting experiments. Figure 2B: pATM, ATM, and pATR bands are unclear, and GAPDH intensity appears to be saturated. Figure 6A: SLFN11/12 intensity appears to be saturated. The quantification of Western blots from triplicate (or more) would be helpful. Additionally, this reviewer wondered why there are two GAPDH bands in Figure 6A/B. Please explain the details.

ANSWER #7:

We have repeated the Western blot experiments to address the concerns. The revised figures now include higher-quality blots obtained with exposures within the linear detection range, resulting in clearer pATM, ATM, pATR and ATR signals (revised **Figure 2B**) as well as distinct SLFN11 and SLFN12 bands (revised new **Extended Data 9 A and C**). We also added quantification data from at least three independent biological replicates, with densitometric values normalized to the corresponding loading controls (new **Extended Data 9 A and C**). Regarding the two GAPDH bands, SLFN11 and SLFN12 were detected on separate membranes, and we therefore displayed the respective loading controls for each. We agree that the original GAPDH

signals were oversaturated; consequently, we repeated the experiments using β -actin as an alternative loading control to prevent signal saturation.

POINT #8

The authors should check whether DDR inhibitors and/or oxidative stress inhibitors downregulate MAPK phosphorylation **under viral infection**.

ANSWER #8:

Published studies have shown that inhibitors of the DNA damage stress response pathway the use of and antioxidant impair coronaviruses infection (**See References 1–4 listed at the end of this response**).

To further validate this point and the link between DDR and oxidative stress, we showed that treatment of HCoV-OC43 infected cells with DDR inhibitors (ATM) reduced virus-induced MAPK phosphorylation (p-p38) and viral protein production. These data are included in the revised figures (**Extended Data 2C**) and the following text has been added to the manuscript at lines 181 - 182:

“Furthermore, inhibitors of the DNA damage stress response reduce both *p-p38 levels and coronavirus protein expression (Extended Data 2C)*.”

References:

1. Liubing Du et al., Oxidative stress transforms 3CLpro into an insoluble and more active form to promote SARS-CoV-2 replication. *Redox Biology*, Volume 48, 2021, 102199, ISSN 2213-2317, <https://doi.org/10.1016/j.redox.2021.102199>.
2. Olgagnier D et al., SARS-CoV2-mediated suppression of NRF2-signaling reveals potent antiviral and anti-inflammatory activity of 4-octyl-itaconate and dimethyl fumarate. *Nat Commun*. 2020 Oct 2;11(1):4938. Correction to: *Nature Communications* <https://doi.org/10.1038/s41467-020-18764-3>
3. Ordonez, A.A., Bullen, C.K., Villabona-Rueda, A.F. et al. Sulforaphane exhibits antiviral activity against pandemic SARS-CoV-2 and seasonal HCoV-OC43 coronaviruses in vitro and in mice. *Commun Biol* 5, 242 (2022). <https://doi.org/10.1038/s42003-022-03189-z>
4. Zhang, S., Wang, J., Wang, L. et al. SARS-CoV-2 virus NSP14 Impairs NRF2/HMOX1 activation by targeting Sirtuin 1. *Cell Mol Immunol* 19, 872–882 (2022). <https://doi.org/10.1038/s41423-022-00887-w>

POINT #9

Homo sapiens should be italicized in Figure 1 legends and Extended Data 1 legends.

ANSWER #9:

Corrected as requested.

POINT #10

The text in the heatmap of Extended Data 6 is too small and needs adjustment for readability.

ANSWER #10:

Corrected as requested (now **Extended Data Figure 10**).

POINT #11

The conclusion sentence in lines 172-174 was too strong from the data. The authors should tone down the nuance.

ANSWER #11:

We have now modified the text adding new data to support the statement, lines 178 -182.

“These findings suggest that coronaviruses have adapted their codon usage to thrive in a tRNA environment in which stress responses are activated. Consistent with this, previous studies have shown that inhibitors of the DNA damage stress response pathway and antioxidants impair coronavirus infection. Furthermore, inhibitors of the DNA damage stress response reduce both p-p38 levels and coronavirus protein expression (Extended Data 2C)”.

POINT #12

The sentence in lines 214-217, the authors explained that f⁵C was found in mt-tRNAs. This was indeed true for Q. This point should be highlighted.

ANSWER #12:

We thank the reviewer for the comment. We have now added the following sentence at lines 230 – 231.

“Recent studies suggest that Q modification, as f⁵C, may also occur in mitochondrial tRNAs.

POINT #13

The authors should discuss manQ and galQ as well.

ANSWER #13:

We have now included the following text in the revised manuscript addressing the potential impact of manQ and galQ on viral codon translation at lines 136 – 139:

“The Q modification and its derivatives, mannosyl-Q (manQ) and galactosyl-Q (galQ), favor the decoding of four U-ending codons (Asparagine (Asn) AAU, Aspartic acid (Asp) GAU, Histidine (His) CAU, and Tyrosine (Tyr) UAU), with galQ found in human tRNA^{Tyr}, and manQ found in human tRNA^{Asp} (Figure 1D)”.

REVIEWER COMMENTS

Reviewer #1 (Remarks to the Author):

POINT #1

The authors have addressed the majority of my concerns, as well as those raised by the other reviewers, and have significantly improved the manuscript. This reviewer greatly appreciates the approach of knocking down relevant modification enzymes to demonstrate a cause-effect-relationship between tRNA modification patterns and viral protein synthesis. The observation that knockdown of ELP3 and QTRT1, as well as overexpression of ADAT2 and ALKBH1, reduces viral protein production is highly interesting. To fully address my concern the authors should make an effort to better visualize the results presented in Figure 4. Quantification of knockdown/overexpression efficiency and the corresponding effects on viral protein levels would greatly help the reader.

ANSWER #1

We redesigned **Figure 4** to improve clarity and interpretability. The revised figure now includes quantification of knockdown and overexpression efficiency for each experiment, shown as bar graphs alongside representative Western blots, as well as quantification of the corresponding effects on viral protein levels.

POINT #2

It should also be noted that some knockdown effects appear quite subtle, raising doubts about their impact on the associated tRNA modifications. I suggest that, for at least one enzyme, the authors provide targeted tRNA modification analysis to demonstrate a reduction in the relevant modifications following knockdown. Some knockdown effects appear subtle; for at least one enzyme, direct evidence of reduced tRNA modification is needed.

ANSWER #2

We now provide direct evidence that knockdown of the relevant enzyme reduces the corresponding tRNA modification. LC-MS/MS analysis shows that ELP3 knockdown in HCoV-OC43 infected A549 cells significantly reduces the mcm⁵U modification.

siRNA-mediated knockdown of *ELP3* (mcm⁵U/ mcm⁵s²U pathway) in HCoV-OC43 (MOI 0.1, 48 hpi) infected A549 cells. HCoV-OC43 N-protein (NP) levels was measured by western blot (A). mcm⁵U tRNA modification levels from the same samples was measured by LC-MS/MS (B). Non targeting siRNA (siRNA-scr) was used as control ($p < 0.05$, unpaired t -test, $n=2$).

Reviewer #2 (Remarks to the Author):

We have completed our review of the revised manuscript. The authors have been responsive to the comments and have adequately addressed the concerns raised in our previous review. The additional experiments/analyses

and clarifications have strengthened the manuscript. We believe the revised version is now suitable for publication.

Reviewer #3 (Remarks to the Author):

POINT #1

This reviewer thanks the authors for their extensive revisions and thoughtful responses to this reviewer's comments. The manuscript has been substantially improved. This reviewer appreciates the extensive efforts involved. The addition of enzyme perturbation experiments strengthens the work. However, this reviewer still has remaining concerns, primarily regarding the ribosome profiling analysis. This reviewer appreciates the addition of enzyme knockdown/overexpression experiments and the codon frequency analysis shown in Extended Data 5. These are valuable data. However, this reviewer still respectfully and strongly requests doing ribosome profiling analysis. The authors attribute their decision to omit ribosome profiling to technical limitations. However, this reviewer finds that this justification is premature. The authors cite a study (Finkel et al., 2021) as evidence that ribosome profiling at late infection stages is technically problematic due to loss of footprint periodicity on viral RNAs. However, another study (Kim et al., 2021), whose dataset the authors themselves use for the codon frequency analysis in Extended Data 5, appears to show clear 3-nucleotide periodicity at 36 hours post-infection. Thus, there is no technical hurdle for doing ribosome profiling in the author's condition, according to the literature. This reviewer recommended doing the new ribosome profiling experiments that matched the authors' tRNA-Seq/mass spec data. Alternatively, the authors could reanalyze the Kim et al. dataset to calculate ribosome occupancy on each codon (viral and host mRNAs) (not the codon frequency analyzed in the Extended Data 5) and test the correspondence with tRNA modification changes (e.g., Q-modified, mcm5U-modified, and I-modified tRNAs). For viral mRNAs, this reviewer understands the authors' concerns about non-ribosomal RNP complexes contributing to background signal, as noted in previous studies (Finkel et al., 2021; Jungfleisch et al., 2022). However, analytical strategies may address this issue. For instance, the authors could filter footprints by length, retaining only fragments in the 28-30 nucleotide range that match the canonical ribosome footprint size observed on host mRNAs. Alternatively, codon-specific metagene profiling, which averages ribosome occupancy around specific codons across many transcript instances, can reveal genuine translation signals even when background noise is present. Comparing such metagene profiles between mock-infected and infected samples (24-36 hpi) for codons affected by Q, mcm5U, and I modifications would directly test whether tRNA modification changes influence ribosome transit. This reviewer encourages the authors to explore whether such approaches could be applied. The authors propose that tRNA-modifying enzymes are regulated post-transcriptionally during infection, based on decreased mRNA but increased protein levels (Figure 3G-J, Extended Data 6). This claim could be directly validated using the Kim et al. (2021) ribosome profiling dataset. Specifically, this reviewer asks the authors to analyze the translation efficiency (TE, calculated as ribosome-protected fragment reads / mRNA reads) of the key tRNA-modifying enzyme mRNAs—ELP3, QTRT1, ADAT2, ALKBH1, and NSUN2/3—across the infection time course to check the consistency with TE results in Extended Data 6.

ANSWER #1

We performed an in-depth reanalysis of the **Kim et al. (2021)** ribosome profiling dataset:

1. Host mRNAs enriched in codons corresponding to infection-induced tRNA modifications display reduced ribosome occupancy, consistent with accelerated translation (**Extended Data Figure 5D-F**).

This sentence as now been added to the manuscript at lines 300 -302:

“Moreover, ribosome occupancy analyses confirmed faster translation elongation at infection-induced tRNA-modification-dependent codons in TA transcripts relative to TR ones (Extended Data 5 D-F)”.

2. Ribosome occupancy of tRNA-modifying enzymes could not be reliably assessed due to insufficient read coverage in the available dataset. However, the discordance between protein and mRNA levels (**Extended Data 6**), together with the clear reduction in viral protein levels observed upon knockdown of these enzymes (**Figure 4**), supports a translation-level activation during infection. In the revised

text, we maintain this interpretation but adopt more cautious wording, using “suggest” rather than “indicate” (line 312).

3. Viral ribosome occupancy cannot be directly compared with mock conditions because viral sequences are absent from mock samples, precluding differential occupancy analyses.

POINT #2

Line 136-139: citation of the paper is missing.

ANSWER #2

The citation has been added to the manuscript.

Point by point response.

Reviewer comment:

The authors answered the remaining concerns of this reviewer. The one note could be for the new sentence of “Moreover, ribosome occupancy analyses confirmed faster translation elongation at infection-induced tRNA-modification–dependent codons in TA transcripts relative to TR ones (Extended Data 5 D-F)” in lines 300 - 302. If this reviewer’s understanding is correct, the analysis in Extended Data 5D-F did not focus on the specific subgroup of mRNAs, but rather global changes in all the mRNAs in the dataset. Thus, the phrase regarding TA transcripts and TR transcripts in the corresponding sentence should be irrelevant. The authors should carefully revisit their own analysis and description of the data. Also, the sentence summarized that all the corresponding codons provided faster elongation. However, this may not apply to some of the codons. The authors should rewrite this part more concretely, not exaggerating the results.

Answer:

We have adjusted the sentence at **lines 301 – 302** that now states:

“Moreover, ribosome occupancy analyses revealed reduced occupancy at these codons in in line with enhanced translation elongation during infection (**Supplementary Figure 5 D-F**)”.